

# Modeling Supercritical CO$_2$ Flow and Mineralization in Reactive Host Rocks with PFLOTRAN v7.0

Michael Nole[1,2], Katherine Muller[1], Glenn Hammond[1], Xiaoliang He[1], and Peter Lichtner[3]

[1]Earth Systems Science Division, Pacific Northwest National Laboratory, Richland, WA, USA
[2]ResFrac Corporation, Denver, CO, USA
[3]OFM Research, Santa Fe, NM, USA

**Correspondence:** Michael Nole (michaelnole@resfrac.com)

**Abstract.** Understanding the flow and reactivity of CO$_2$ injected into geologic reservoirs is important for many subsurface applications including secure geologic carbon storage (GCS), critical mineral extraction, enhanced geothermal systems (EGS), and enhanced oil recovery (EOR). Traditionally, subsurface CO$_2$ injection for GCS applications has focused on geologic formations with favorable subsurface configurations for CO$_2$ migration and trapping through non-reactive mechanisms such as
structural, solubility, and petrophysical trapping to isolate CO$_2$ in the subsurface. Recently, CO$_2$-reactive rocks such as mafic and ultramafic basalts have been investigated for their potential to react with injected CO$_2$ in situ to simultaneously dissolve host rock minerals and mineralize CO$_2$ as carbonates. Engineering rapid CO$_2$ mineralization in the subsurface is attractive because of the increased density of stored CO$_2$, the additional safety factors associated with solidification, and the potential to extract valuable critical minerals. However, the limited availability of tools that are capable of modeling the associated coupled
multiphase flow and reactive transport processes, especially at scale, makes it difficult to predict the long term behavior of a commercial-scale CO$_2$ injection into a reactive host rock. Here we present recent developments in the parallel flow and reactive transport simulator PFLOTRAN to model coupled CO$_2$-brine flow and reactive transport for a wide range of injection and production applications involving reactive CO$_2$-brine systems. These developments are based on the well established and trusted CO$_2$ flow capabilities in the STOMP-CO2 simulator. New capabilities added to PFLOTRAN include new CO$_2$-brine
equations of state with optional thermal coupling, several new constitutive relationships like capillary pressure smoothing and scanning path hysteresis, a fully implicit well model, and native linkage with PFLOTRAN's well-established reactive transport libraries. A series of numerical benchmarks between PFLOTRAN and STOMP-CO2 verify the newly developed CO$_2$-brine flow capabilities, and demonstrations of coupled CO$_2$-brine flow modeling and reactive transport show how CO$_2$ mineralization can be engineered in reactive host rocks. Finally, an example use case involving copper leaching by CO$_2$ and critical mineral
extraction is presented to showcase the strengths of this new implementation. Several limitations still remain, including limited availability of field data to parameterize models. Future work should constrain the evolution of mineral surface area during mineralization and the temperature/pH dependence of geochemical reactions for specific systems of interest.





## 1 Introduction

The fate of $CO_2$ injected into the subsurface has been of interest for several decades because of its potential utility to a variety

of subsurface energy engineering challenges, including enhanced oil recovery and geologic carbon storage. More recently, $CO_2$ has been proposed as a potential working fluid for extracting heat from geothermal energy reservoirs using novel well configurations and for producing heat from enhanced geothermal systems (Pruess, 2008; Wu and Li, 2020). It has also been shown that in highly $CO_2$-reactive reservoirs composed of, e.g., mafic and ultramafic basalts, $CO_2$ can mineralize as carbonates on much shorter timescales than in traditional non-reactive reservoirs (McGrail et al., 2017; Pogge von Strandmann et al.,

2019). The geochemistry of these reactions is also such that the dissolution of the host rock that is promoted by enhanced brine acidity could release valuable critical minerals; $CO_2$ has therefore been proposed as a working fluid to promote critical mineral recovery with the added benefit of storing $CO_2$ (Stanfield et al., 2024).

To predict the transient behavior of $CO_2$ injected into the subsurface for any of these applications and to engineer optimal injection strategies, reservoir simulation tools are required. Such tools have matured over the past few decades; recently,

the Society of Petroleum Engineers (SPE) concluded its 11th international Comparative Solution Project, which benchmarked state-of-the-art industry and research simulators on a series of $CO_2$ storage challenge problems in 2D and 3D (Nordbotten et al., 2024). However, many of these tools are limited in their capacity to model coupled multiphase flow and reactive transport in reactive host reservoirs. Lacking these capabilities makes it difficult to predict the long term behavior of a commercial-scale $CO_2$ injection into a reactive host rock, let alone model enhanced critical mineral dissolution and extraction using $CO_2$ as the

working fluid.

The parallel flow and reactive transport simulator PFLOTRAN has been extended to model coupled $CO_2$-brine flow and reactive transport for a wide range of injection and production applications involving reactive $CO_2$-brine-mineral systems. These developments are based on the well established and trusted $CO_2$-brine flow capabilities in the STOMP-CO2 simulator. New capabilities added to PFLOTRAN include new $CO_2$-brine equations of state with optional thermal coupling, several new

constitutive relationships like capillary pressure smoothing and scanning path hysteresis, a fully implicit well model, and native linkage with PFLOTRAN's well-established reactive transport libraries. We first present the theory behind all of this newly-implemented functionality. Then we discuss how to invoke these capabilities in a PFLOTRAN input deck, and we present a suite of benchmark exercises and verification tests to demonstrate how our implementation compares to other well-established simulators like STOMP-CO2 and TOUGH2. Finally, we demonstrate the capabilities on a copper extraction problem using

$CO_2$ to induce copper leaching for critical mineral extraction.

## 2 Theory

Simulating $CO_2$ injection into reactive subsurface reservoirs requires solving systems of nonlinear partial differential equations describing the transport of water, $CO_2$, and salt; heat transfer; and chemical reactions through porous media at large scales. These equations are strongly coupled: for instance, the dissolved concentration of $CO_2$ affects liquid water density,

viscosity, enthalpy and pH among other state variables, which in turn affect the transport and reactivity of other species. In





PFLOTRAN, SCO2 Mode has been developed to fully implicitly solve a set of "flow" governing equations. Those equations include conservation of water mass, conservation of $CO_2$ mass, conservation of salt mass, conservation of energy, and well mass flux conservation. Water, $CO_2$, and salt components partition between three phases (liquid [aqueous], gas [$CO_2$-rich], and salt precipitate); water is miscible in the $CO_2$ phase, and $CO_2$ and salt are miscible in the aqueous phase. Transport of mass in the SCO2 flow mode occurs as the result of pressure gradients within each phase, concentration gradients within each phase, buoyancy, and component sources/sinks. These equations are all solved fully implicitly within SCO2 Mode; some equations, like the energy equation and the well equation, are optional. The governing partial differential equations, constitutive relationships, equations of state, and fully implicit well model formulation were designed with base functionality that emulates the STOMP-CO2 simulator (White et al., 2012) and expands beyond those capabilities.

Modeling transport and reaction of dissolved aqueous species also requires solving a coupled system of mass action equations. We refer to this as the "reactive transport" step. To model multicomponent geochemistry in addition to multiphase $CO_2$-brine flow, PFLOTRAN's SCO2 flow mode has been designed to sequentially couple to PFLOTRAN's global implicit reactive transport (GIRT) mode, a mature and comprehensive reactive transport code (Hammond et al., 2014). SCO2 Mode first solves its set of governing equations for mass and energy fluxes over a given time step. Then, it passes certain variables like Darcy fluxes, porosity, and gas phase saturation to reactive transport. Reactive transport then takes as many sub-steps as are required to complete one "flow" step. If $CO_2$ is produced or consumed through mineralization reactions in reactive transport, transport passes a source/sink term of $CO_2$ back to flow. Flow and reactive transport hand this information off at every flow time step for the duration of the simulation.

## 2.1 Flow Governing Equations

PFLOTRAN's SCO2 Mode solves a fully coupled system of three component mass conservation equations and one energy conservation equation. The energy conservation equation can be optionally disabled in favor of isothermal simulations at user-defined initial temperature conditions (including temperatures read from a restart file). Additionally, an arbitrary number of fully coupled well mass conservation equations can be solved corresponding to each well in the domain with or without thermal coupling. The default mass and energy conservation equations consider conservation of water, $CO_2$, salt, and system internal energy. The equations take the following form:

$$\frac{\partial \phi \left(s_l \rho_l X_w^l + s_g \rho_g X_w^g\right)}{\partial t} = -\boldsymbol{\nabla} \cdot \left(\rho_l X_w^l \boldsymbol{q}_l + \rho_g X_w^g \boldsymbol{q}_g + \boldsymbol{J}_w^l + \boldsymbol{J}_w^g\right) + q_w \tag{1}$$

$$\frac{\partial \phi \left(s_l \rho_l X_{CO_2}^l + s_g \rho_g X_{CO_2}^g\right)}{\partial t} = -\boldsymbol{\nabla} \cdot \left(\rho_l X_{CO_2}^l \boldsymbol{q}_l + \rho_g X_{CO_2}^g \boldsymbol{q}_g + \boldsymbol{J}_{CO_2}^l + \boldsymbol{J}_{CO_2}^g\right) + q_{CO_2} \tag{2}$$

$$\frac{\partial \phi \left(s_l \rho_l X_{salt}^l + s_s \rho_s\right)}{\partial t} = -\boldsymbol{\nabla} \cdot \left(\rho_l X_{salt}^l \boldsymbol{q}_l + \boldsymbol{J}_{salt}^l\right) + q_{salt} \tag{3}$$

$$\frac{\partial \phi \left(s_\alpha \rho_\alpha U_\alpha\right) + \left(1 - \phi\right) C_p^{rock} \rho_{rock} T}{\partial t} = -\boldsymbol{\nabla} \cdot \left(\rho_l H_l \boldsymbol{q}_l + \rho_g H_g \boldsymbol{q}_g - \kappa_{eff} \nabla T\right) + q_e \tag{4}$$





where $\phi$ is porosity, $s_\alpha$ is the saturation of phase $\alpha$, $\rho_\alpha$ is the mass density of phase $\alpha$, $X_\beta^\alpha$ is the mass fraction of component $\beta$ in phase $\alpha$, $q_\alpha$ is the Darcy flux of phase $\alpha$, $J_\beta^\alpha$ is the diffusive flux of component $\beta$ in phase $\alpha$, $q_\beta$ are the sources and sinks of component $\beta$, $U_\alpha$ is the internal energy of phase $\alpha$, $C_p^{\text{rock}}$ is the heat capacity of the rock, $\rho_{\text{rock}}$ is the solid rock density, $T$ is temperature, $H_\alpha$ is the enthalpy of phase $\alpha$, $\kappa_{\text{eff}}$ is the effective thermal conductivity of the medium, and $q_e$ are the energy sources and sinks.

Three phases are considered in this formulation: liquid phase, $CO_2$-rich phase, and salt precipitate phase. Within the $CO_2$-rich phase, trapped and free-phase saturations are tracked separately for performing optional hysteresis calculations. The salt precipitate phase contains only the salt component; the salt component can also partition into the aqueous phase, but it cannot exist in the $CO_2$-rich phase. The state of the $CO_2$-rich phase can be either liquid, gaseous, or supercritical $CO_2$. For convenience, the $CO_2$-rich phase will be labeled the "gas" phase throughout this document. This phase can contain both $CO_2$ and 95 water components.

## 2.2 Constitutive Relationships

Advective fluxes of the mobile phases are modeled using Darcy's equation:

$$\boldsymbol{q}_\alpha = -\frac{kk_{r_\alpha}}{\mu_\alpha}\nabla\left(p_\alpha - \gamma_\alpha z\right) \tag{5}$$

where $k$ is the permeability of the medium, $k_{r\alpha}$ is the relative permeability of phase $\alpha$, $\mu_\alpha$ is the viscosity of phase $\alpha$, $p_\alpha$ is the 100 pressure of phase $\alpha$, $\gamma_\alpha$ is the specific gravity of phase $\alpha$, and $z$ is vertical elevation.

Relative permeability is computed as a function of pore fluid saturations through one of several available relative permeability function options. Please see the PFLOTRAN Documentation for a comprehensive list of available relative permeability functions for liquid and gas phases. When considering effects of hysteresis on gas trapping, SCO2 Mode removes trapped gas from the mobile gas phase saturation used in gas relative permeability computations.

Diffusive fluxes in the aqueous phase are modeled using Fick's Law:

$$\boldsymbol{J}_{\text{CO}_2}^{\text{l}} = D_{\text{CO}_2}^{\text{l}}\nabla c_{\text{CO}_2} \tag{6}$$

$$\boldsymbol{J}_{\text{salt}}^{\text{l}} = D_{\text{salt}}^{\text{l}}\nabla c_{\text{salt}} \tag{7}$$

$$\boldsymbol{J}_{\text{w}}^{\text{l}} = 1 - \boldsymbol{J}_{\text{CO}_2}^{\text{l}} - \boldsymbol{J}_{\text{salt}}^{\text{l}} \tag{8}$$

where $D_\beta^l$ is the diffusivity of component $\beta$ in the liquid and $\nabla c_\beta$ is the gradient in dissolved concentration of component $\beta$. 110 Transport of dissolved salt through the aqueous phase is additionally weighted using the Patankar method (Patankar, 2018).

Likewise, gas phase diffusive flux is modeled as follows:

$$\boldsymbol{J}_{\text{CO}_2}^{\text{g}} = D_{\text{CO}_2}^{\text{g}}\nabla c_{\text{CO}_2} \tag{9}$$

$$\boldsymbol{J}_{\text{w}}^{\text{g}} = 1 - \boldsymbol{J}_{\text{CO}_2}^{\text{g}} \tag{10}$$

where $D_\beta^g$ is the diffusivity of component $\beta$ in the gas, $\nabla c_\beta$ is the gradient in mass fraction of component $\beta$ in the gas, and the 115 salt component is not present in the gas phase.





### 2.3 Equations of State

Several options exist in PFLOTRAN to customize the set of equations of state (EOS) used in a simulation to fit a desired application. Here we describe the default options available in SCO2 Mode. We refer the user to the PFLOTRAN Documentation for more information on EOS options.

#### 2.3.1 Density

With the exception of the salt precipitate phase, the composite density of each phase is computed by correcting the pure phase density as a function of component concentrations. The salt precipitate is considered to be a pure "salt" component phase. The default density equation for salt is a function of temperature and pressure and assumes that salt is composed entirely of NaCl (Battistelli et al., 1997).

For the aqueous liquid (non $CO_2$-rich) phase, the density of pure water is first computed as a function of pressure and temperature (Meyer et al., 1993). Brine density is then computed as a function of pure water density and salt mass fraction (Phillips et al., 1981). Finally, the composite mixture density is calculated as a function of pure water density, brine density, and mass fractions of salt and $CO_2$ components (Alendal and Drange, 2001).

In the gas phase, a similar approach is taken with the exception that there is no dissolved salt in the gas phase. Pure gas phase density (Span and Wagner, 1996) and water vapor density (Meyer et al., 1993) are first calculated as functions of pressure and temperature. Then, the gas mixture density is computed as a weighted average of the two densities weighted by gas phase component mass fractions. Pure $CO_2$ thermodynamic properties (density, viscosity, internal energy, and enthalpy) can either be computed or read in through a database for greater efficiency. A generic $CO_2$ thermodynamic property table based off of the Span-Wagner EOS is included in the PFLOTRAN database directory and it is recommended to use this lookup table for $CO_2$ simulations.

#### 2.3.2 Vapor Pressure

The liquid vapor pressure is computed as a function of temperature, salinity, and capillary pressure. Pure water saturation pressure is first computed as a function of temperature (Meyer et al., 1993), and then an adjustment is applied to include the effect of salinity (Haas, 1976). The Kelvin equation is then applied to calculate the reduced vapor pressure as a function of capillarity (Nitao, 1988).

#### 2.3.3 Viscosity

Similar to density, the viscosity of each mobile phase is computed as a function of temperature, pressure, and phase composition. For the liquid phase, the viscosity of pure water is computed as a function of temperature and pressure (Meyer et al., 1993). This pure water viscosity is then adjusted as a function of temperature and salinity (Phillips et al., 1981) to get the brine viscosity. The viscosity of $CO_2$ is separately calculated as a function of temperature and density (Fenghour et al., 1998), and the final composite liquid viscosity is computed as a function of $CO_2$ mass fraction, $CO_2$ viscosity, and brine viscosity.





In the gas phase, viscosity is calculated as a function of pure water viscosity, pure $CO_2$ viscosity, and mass fractions of each component in the gas phase (Poling et al., 2001).

### 2.3.4 Diffusion Coefficients

Diffusion drives fluxes of dissolved $CO_2$ and dissolved salt in the aqueous phase, and it drives water vapor diffusion in the gas phase. Diffusive flux is governed by concentration gradients and molecular diffusion coefficients of each component in each phase. In the aqueous phase, the diffusion coefficient of $CO_2$ is computed as a function of temperature (Cadogan et al., 2014) and dissolved salt mass fraction (Belgodere et al., 2015). The molecular diffusion coefficient of salt in the aqueous phase is computed using the Gordon method (Poling et al., 2001) and as a function of mean ionic activity (Bromley, 1973). The
diffusivity of water vapor in the gas phase is computed as a function of temperature and pressure via the method of Wilke and Lee (Poling et al., 2001).

### 2.3.5 Two-Phase Equilibrium

When dissolved $CO_2$ exceeds its solubility in the aqueous phase, or when water vapor condenses from the $CO_2$-rich phase, a two-phase state can arise in a model. That is, in a given cell/location, both the liquid (water-rich) and gas ($CO_2$-rich) phases can
coexist. In SCO2 Mode, the transition from single- to two-phase is treated as an equilibrium process. Two-phase coexistence therefore requires an equilibrium partitioning model to determine how $CO_2$ and water partition between phases at a given pressure, temperature, and salinity. This equilibrium partitioning is used to determine the solubility of $CO_2$ dissolved in water and the partial pressure of water vapor in the gas; these are used as conditions for determining the transition between the single-phase aqueous state and the two-phase state or the single phase gas state and the two phase state, respectively. At temperatures
below $100^{o}$C, the method of (Spycher et al., 2003) is used. Above $100^{o}$C, the method of (Spycher and Pruess, 2010) with salinity correction is used.

### 2.3.6 Internal Energy and Enthalpy

Enthalpy of the liquid phase is computed as a function of temperature, pressure, dissolved $CO_2$ concentration, dissolved salt concentration, and individual pure component enthalpies (Battistelli et al., 1997). Pure liquid water enthalpy is computed from
the pure water EOS (Meyer et al., 1993) and then adjusted as a function of salinity (Michaelides, 1981). Pure $CO_2$ enthalpy is computed from the $CO_2$ EOS (Span and Wagner, 1996). In the gas phase, the composite gas phase enthalpy is computed as a weighted average of water vapor enthalpy and pure $CO_2$ enthalpy, weighted by the mass fraction of each component in the gas. Finally, the internal energy of each phase is computed by taking phase enthalpy and subtracting cell pressure divided by phase density.





## 2.4 Fluid-Rock Properties


Several fluid-rock properties are computed as functions of the state of the pore fluids in the system. Separate gas phase and liquid phase tortuosities are computed as functions of rock porosity, liquid saturation, and gas saturation (Millington and Quirk, 1959). When dissolved salt mass becomes supersaturated in the aqueous phase, it precipitates as a solid and changes both the effective porosity and the absolute permeability of the medium (Verma and Pruess, 1988). Relative permeability of each mobile phase is computed as a function of phase saturations; when modeling trapped gas hysteresis, the liquid-saturated end point of the gas phase relative permeability model adapts as a function of trapped gas saturation. The $CO_2$-brine-rock interfacial tension is computed as a function of salt concentration and temperature (Abramzon and Gaukhberg, 1994). This results in a decrease in capillary pressure with decreasing interfacial tension.


### 2.4.1 Unsaturated Capillary Pressure Extensions


At the unsaturated end of a typical capillary pressure function (e.g., Van Genuchten or Brooks-Corey), there can exist a non-zero residual liquid saturation toward which the capillary pressure increases asymptotically. This is due to the fact that drainage of the wetting (aqueous) phase by the non-wetting (gas) phase is limited by the connectivity of the liquid phase in the porous medium. When the liquid phase is completely disconnected, the gas phase is unable to displace any more liquid and therefore capillary pressure can theoretically become infinite. However, under miscible two-phase flow conditions where both the water and the $CO_2$ components can dissolve into both phases, it is possible to drive water saturation below the residual saturation without displacing fluid. In the case of dry $CO_2$ injection, continuous pumping of a pure $CO_2$ phase into a brine-saturated reservoir can cause desaturation of the rock through desiccation, where water molecules evaporate from the liquid phase into the free $CO_2$ phase.


Capturing this phenomenon requires extending the capillary pressure past the residual liquid saturation. Using the approach of Webb (2000), the capillary pressure function is divided into two regimes: above and below the matching point aqueous saturation. The matching point saturation is the point at which the capillary pressure curve transitions between its original form and its extension to low aqueous saturation conditions. The matching point is computed by finding the aqueous saturation at which the slope of the original capillary pressure curve matches that of the extended curve given a specified capillary pressure at zero liquid saturation (i.e., the oven-dried capillary pressure) and a specified extension shape. In PFLOTRAN, several types of unsaturated extensions are available for the standard Van Genuchten capillary pressure model. Additionally, this work developed a capillary head-based Van Genuchten model with a logarithmic extension (Fayer and Simmons, 1995) and a Brooks-Corey capillary pressure with a logarithmic extension, both of which are compatible with the scanning path hysteresis model.



### 2.4.2 Gas Trapping Hysteresis


Gas trapping occurs when an aqueous phase imbibes through a medium that is partially saturated with a gas phase. The gas phase can become disconnected and therefore immobilized (though, analogous to the desiccation process, $CO_2$ can continue to





dissolve into water and diffuse through the aqueous phase) as aqueous phase saturation increases throughout the medium. The extent to which the free gas phase can be trapped depends on the degree to which the medium was saturated with gas prior to imbibition. The process of gas trapping can therefore be formulated as a hysteretic phenomenon via a scanning path hysteresis

model (Parker and Lenhard, 1987). In PFLOTRAN, the simplified Parker and Lenhard model is implemented (Kaluarachchi and Parker, 1992). This model is compatible with the Brooks-Corey capillary pressure model and the capillary head-based Van Genuchten model, as well as any choice of relative permeability functions.

## 2.5 Fully Implicit Well Model

An optional fully coupled well model can inject $CO_2$ or produce reservoir fluids in user-specified portions of a model domain

and dynamically adapt to changes in pressure along the wellbore, pressure in the reservoir, and physical properties of the formation. The well model is solved on an embedded sub-domain of the reservoir grid; where a well segment passes through a reservoir cell, the associated flux from well to reservoir is incorporated into the residual and Jacobian calculations.

To construct a well, the well trajectory is first defined by the user in terms of line segments and curves. The wellbore is then discretized as a function of the reservoir grid discretization, and finally individual well segments are each coupled to their

corresponding reservoir grid cells.




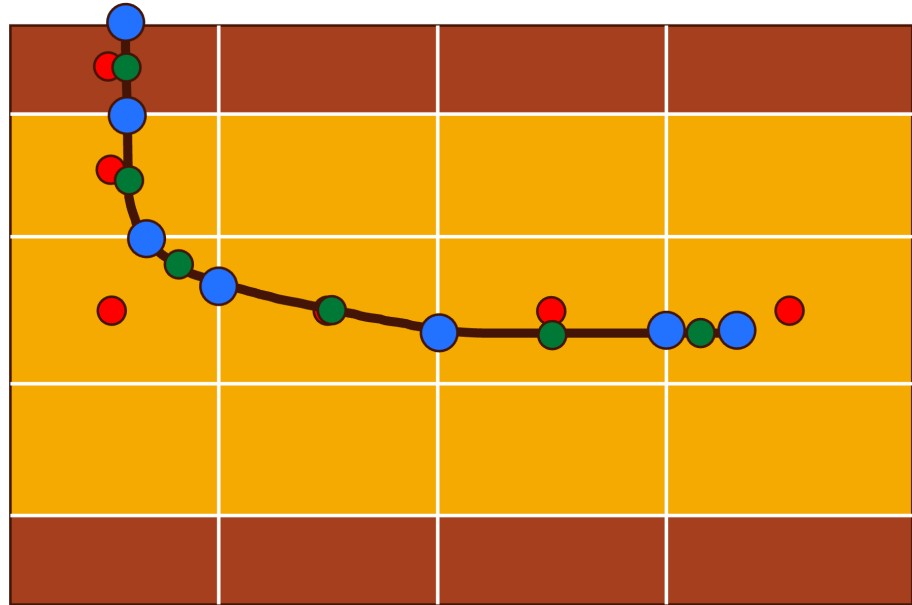

**Figure 1.** Discretized well model embedded in a layered reservoir. Blue dots mark the ends of each well segment, green dots mark well segment centers, and red dots mark reservoir cell centers.

When embedded into a discrete reservoir, a well model is defined in terms of well segments, well segment centers, and the reservoir cell center through which each well segment passes. In Figure 1, well segments are defined by the intervals between blue circles. Green circles denote the well segment centers, which are defined by their centroids. Red circles denote the center of each reservoir cell through which the well passes.

Invoking the well model adds one extra equation to the set of governing equations in the following form:

$$\sum_i Q_{w,\alpha}^i = q_{w,\alpha} \tag{11}$$

where $Q_w$ is the mass flow rate of phase $\alpha$ between the reservoir and the well in well segment $i$, and $q_w$ is the top hole mass flow rate of phase $\alpha$ into or out of the well. The fluid flux in or out of the well at a given well segment is computed as a function of the pressure difference between the center of each well segment and the reservoir cell that the segment occupies, where the

cell-centered reservoir pressure is adjusted by a hydrostatic correction to the same depth as the well segment:

$$Q_{w,\alpha}^i = -\frac{WI\rho_\alpha}{\mu_\alpha}\left(P_w^i - (P_r^i + \rho_\alpha \boldsymbol{g} z_{w-r})\right) \tag{12}$$

where $\rho_\alpha$ is the density of phase $\alpha$, $\mu_\alpha$ is the viscosity of phase $\alpha$, $P_w$ is the wellbore pressure at the center of well segment $i$, $P_r^i$ is the pressure in the reservoir cell occupied by well segment $i$, $g$ is the gravity vector, and $z_{w-r}$ is the vertical offset between well cell center and reservoir cell center. The well index, $WI$, is computed as a function of the directional permeability





of the reservoir in addition to well properties like well orientation, skin factor, casing, and wellbore radius through a modified 3D anisotropic Peaceman equation (Peaceman, 1978, 1983) using the projection of an arbitrarily oriented well segment onto the principal axes of the domain (Shu, 2005; White et al., 2013).

In the fully implicit formulation, bottomhole pressure is solved as a primary solution variable in addition to the reservoir primary variables. Wellbore pressure in each segment is then computed as a function of well bottomhole pressure by assuming hydrostatic pressure conditions along the well. Fluid densities in each segment of the well are computed as functions of pressure and temperature; the elevations of well segment centers are fixed at the beginning of a simulation.

Well pressure control can be imposed individually upon each well either through a specified fracture pressure (injection wells) or through a minimum pressure (extraction wells). Wells are initialized as rate-controlled until the well pressure hits either the fracture pressure or the minimum pressure. If one of these pressure bounds is exceeded, the well switches to pressure controlled, whereby the pressure in the well is fixed and flow rates are computed accordingly. This can be used to prevent either injection wells from injecting at pressures that would fracture the host rock or extraction wells from generating non-physical suction. Note that when a well is pressure-controlled, Equation 11 will not be satisfied until the well returns to rate-controlled (i.e., the well pressure falls within the user-specified bounds again).

## 2.6 Primary Variables and Phase States

In addition to modeling miscible coupled flow of $CO_2$-rich and aqueous phases, SCO2 Mode also models phase appearance and disappearance. Depending on the phase state of the system, PFLOTRAN solves its system of governing equations for different primary variables. When phase state changes, those primary variables are also changed accordingly. For example, if a grid cell begins a simulation in the aqueous state, only liquid water, dissolved $CO_2$, and dissolved salt exist in that grid cell. In this state, the primary variables are liquid pressure, $CO_2$ mass fraction, salt mass fraction per unit mass brine, and (optionally) temperature. If $CO_2$ is injected into the cell, dissolved $CO_2$ mass fraction increases until it exceeds the solubility of $CO_2$. Once this happens, dissolved $CO_2$ mass fraction becomes fixed and another primary variable must be used to solve the system of equations. The cell therefore transitions to the two-phase state and updates its primary variables. The transition from aqueous to two-phase state requires only one primary variable to change: $CO_2$ mass fraction to gas phase pressure. SCO2 Mode considers four phase states: an aqueous state with no free-phase $CO_2$, a two-phase state with both aqueous and free-phase $CO_2$, a gas state with only free-phase $CO_2$, and a trapped gas state with only aqueous liquid phase and immobile trapped gas phase (Table 1). When designing a model, care should be taken to identify which phase state the grid cells should be assigned at initialization and choose the set of primary variable constraints accordingly (see Section 3.2 for more discussion).



**Table 1.** Primary Variables and Phase States

| Phase State | Primary Variables |
|---|---|
| Aqueous State | $P_l, X^l_{CO_2}, m^b_{NaCl}, T^*.$ |
| Two-Phase State | $P_l, P_g, m^b_{NaCl}, T^*.$ |
| Gas State | $P_g, P_{CO_2}, M^b_{NaCl}, T^*.$ |
| Trapped Gas State | $P_l, s_{tg}, m^b_{NaCl}, T^*.$ |

Temperature is only required for thermal models.

### 2.7 Reactive Transport Governing Equations

PFLOTRAN's reactive transport process model couples multiphase, multicomponent transport and biogeochemical reaction
through the governing mass conservation equation:

$$\sum_\alpha \left( \frac{\partial \phi s_\alpha \psi^\alpha_j}{\partial t} + \boldsymbol{\nabla} \cdot \left( \boldsymbol{q}_\alpha - \phi s_\alpha \tau_\alpha \boldsymbol{D}_\alpha \nabla \right) \psi^\alpha_j \right) = q_j - \sum_r \nu_{j,r} I_r \tag{13}$$

with porosity $\phi$, saturation $s$, total component concentration $\psi$, Darcy velocity $\boldsymbol{q}$, tortuosity $\tau$, hydrodynamic dispersion tensor
$\boldsymbol{D}$, and source/sink term $q$ for primary species $j$ in phase $\alpha$. Kinetic rates are scaled by the stoichiometry of species $j$ in each
reaction $r$. The total component concentrations for the liquid and gas phases are defined, respectively, as

$$\psi^l_j = c_j + \sum_i \nu_{ji} \chi_i, \tag{14}$$

$$\psi^g_j = \sum_{i'} \nu^g_{ji'} C^g_{i'}. \tag{15}$$

with aqueous free ion concentration $c$, secondary aqueous complex concentration $\chi$, and gas concentration $C^g$. $\nu_{ji}$ and $\nu^g_{ji'}$ are
the stoichiometric contribution of the free ion species $j$ to each secondary aqueous complex and gas.

Secondary aqueous complex and gas concentrations are defined, respectively, by the expressions

$$\chi_i = \frac{K_i}{\gamma_i} \prod_j (\gamma_j c_j)^{\nu_{ji}}, \tag{16}$$

$$C^g_{i'} = \frac{K^g_{i'}}{\gamma_{i'}} \prod_j (\gamma_j c_j)^{\nu_{ji'}} \tag{17}$$

through the law of mass action (Lichtner, 1985).

### 2.7.1 Solubility of CO$_2$ in the Aqueous Phase

Following Duan and Sun (2003), equilibrium is defined as equality of the chemical potential ($\mu$) of CO$_2$ in liquid and gas
phases

$$\mu^l_{CO_2} = \mu^g_{CO_2}. \tag{18}$$





For multi-component systems where $CO_{2(aq)}$ may not be the primary species for the $CO_2$ component (e.g., use of $HCO_3^-$ instead of $CO_{2(aq)}$), equilibrium is generalized to

$$\sum_j \nu_{ji'} \mu_j^l = \mu_{i'}^g, \tag{19}$$

where the sum $j$ is over all primary species and $\nu_{ji'}$ is the stoichiometric contribution of each primary free ion species to the secondary $CO_{2(aq)}$ species. Substituting the expressions for the chemical potentials equating liquid and gas potentials

$$\mu_j^l = \mu_j^{l(0)} + RT \ln \gamma_j m_j, \tag{20}$$

$$\mu_{i'}^g = \mu_{i'}^{g(0)} + RT \ln \varphi_{i'} p_{i'}^g, \tag{21}$$

where $\varphi_{i'}$ and $p_{i'}^g$ are respectively the fugacity coefficient and partial pressure for species $i'$ in the gas phase. Substituting Eqs. 20–21 into Eq. 19 yields

$$\sum_j \nu_{ji'}^g \left( \mu_j^{l(0)} + RT \ln \gamma_j m_j \right) = \mu_{i'}^{g(0)} + RT \ln \varphi_{i'} p_{i'}^g, \tag{22}$$

and alternatively,

$$-\frac{1}{RT} \left( \mu_{i'}^{g(0)} - \sum_j \nu_{ji'}^g \mu_j^{l(0)} \right) = \ln \varphi_{i'} p_{i'}^g - \sum_j \nu_{ji'}^g \ln \gamma_j m_j. \tag{23}$$

Or in terms the gas equilibrium constant $K_{i'}^g$

$$K_{i'}^g = \frac{\varphi_{i'} p_{i'}^g}{Q_{i'}^g}, \tag{24}$$

with the equilibrium constant related to the standard state chemical potentials by the equation

$$\ln K_{i'}^g = -\frac{1}{RT} \left( \mu_{i'}^{g(0)} - \sum_j \nu_{ji'}^g \mu_j^{l(0)} \right) \tag{25}$$

and

$$Q_{i'}^g = \prod_j (\gamma_j m_j)^{\nu_{ji'}^g}. \tag{26}$$

Substituting Eq. 26 and the exponential of Eq. 25 into Eq. 24 results in

$$\frac{\varphi_{ig} p_{i'}^g}{\prod_j (\gamma_j m_j)^{\nu_{ji'}^g}} = \exp \left( -\frac{1}{RT} \left( \mu_{i'}^{g(0)} - \sum_j \nu_{ji'}^g \mu_j^{l(0)} \right) \right) \tag{27}$$





### 2.7.2 Contribution to CO$_2$ Residual Equation

For grid cells with a nonzero CO$_2$ gas saturation (i.e., $s_g > 0$), the governing mass conservation equation (i.e., Eq. 13) for CO$_2$

is replaced by a residual equation that constrains the CO$_{2(aq)}$ concentration against the solubility of CO$_2$. The residual equation

for the primary CO$_2$ species (i.e., CO$_{2(aq)}$, HCO$_3^-$ or CO$_3^{-2}$) is then based on Eq. 27, i.e.,

$$f_{CO_2} = \prod_j (\gamma_j m_j)^{\nu^g_{jCO_2}} - \frac{\varphi_{CO_2} p^g_{CO_2}}{\exp\left(-\frac{1}{RT}\left(\mu^{g(0)}_{CO_2} - \sum_j \nu^g_{jCO_2}\mu^{1(0)}_j\right)\right)} = 0 \tag{28}$$

with Jacobian entries

$$\frac{\partial f_{CO_2}}{\partial m_{j'}} = \frac{\partial \left(\prod_j (\gamma_j m_j)^{\nu^g_{jCO_2}}\right)}{\partial m_{j'}} \quad \forall \, j' \tag{29}$$

where $f_{CO_2}$ is the residual equation for the CO$_2$ component and $j'$ represents the primary species.





## 3 Usage

The following section describes new input parameters specific to SCO2 Mode, the well model, and coupling with reactive transport. All other base PFLOTRAN functionality (e.g., gridding, material property definition, output options, dataset I/O, etc) can be used with SCO2 Mode just like all other flow modes (see the PFLOTRAN Documentation for more details).

Running models with SCO2 Mode requires invoking the flow mode through the input deck and then choosing from a variety of input options specific to SCO2 Mode. Several new options have been added to the input deck that allow the user to choose the functionality they desire and parameterize accordingly. The following sections provide the user with a description of new input options, associated keywords, and snapshots of input decks providing context on where those keywords are used.

### 3.1 New Inputs

The two highest-level input blocks required to run a subsurface model with SCO2 Mode are the SIMULATION and SUB-SURFACE blocks. In the SIMULATION block, the user chooses which process models they would like to simulate. Those can include any combination of flow modes, transport modes, geomechanics, well models, and more. PFLOTRAN will couple these processes together according to their default coupling schemes and provide coupling options where available. To select SCO2 Mode, the user would invoke MODE SCO2 under SUBSURFACE_FLOW within the PROCESS_MODELS sub-block
of SIMULATION.

Additionally, an OPTIONS block within SUBSURFACE_FLOW can be used to define certain refinements to the process model. Most notably, this is where the user would decide whether to run an isothermal simulation or a non-isothermal simulation. To run isothermal, the user has two options: first, ISOTHERMAL_TEMPERATURE <temperature value> sets a single constant temperature equal to <temperature value> throughout the entire model domain. Second, the user could invoke
FIXED_TEMPERATURE_GRADIENT to specify initial temperature conditions in the model and keep those temperatures fixed over time. Use of the FIXED_TEMPERATURE_GRADIENT keyword requires including thermal state variables in the FLOW_CONDITIONs and then setting cell temperatures through an INITIAL_CONDITION.

Other options in the OPTIONS block include:

- UPDATE_SURFACE_TENSION scales the $CO_2$-brine-rock interfacial tension as a function of temperature and brine
saturation, which in turn affects capillary pressure. Otherwise, no interfacial tension scaling is used.

- MIXTURE_DENSITY provides options for computing $CO_2$-brine mixture density.

- UPWIND_VISCOSITY fully upwinds the viscosity for use in flux calculations.

- PHASE_PARTITIONING can select the $CO_2$-brine equilibrium phase partitioning calculation method.

- NO_H2O_SOURCE_UPDATE_FROM_TRANS turns off water source/sink due to mineralization reactions if reactive
transport is being used



Another option in the PROCESS_MODELS block is to invoke the WELL_MODEL. The WELL_MODEL then expects a TYPE. Currently, TYPE HYDROSTATIC is supported by SCO2 Mode, and the well model coupling strategy is fully implicit. The PROCESS_MODELS block is also where a user would request the SUBSURFACE_TRANSPORT for coupling flow, a well model, and/or reactive geochemical transport. A list of new keywords can be found in Table 2.

**Table 2.** Input Deck Keywords.

| Keyword(s) | Parent Block(s) | Description |
|---|---|---|
| MODE SCO2 | SUBSURFACE_FLOW | Invokes the SCO2 flow mode. |
| WELL_MODEL | SUBSURFACE_FLOW | Invokes the well model. |
| ISOTHERMAL_TEMPERATURE | MODE SCO2, OPTIONS | Sets a single model temperature. |
| FIXED_TEMPERATURE_GRADIENT | MODE SCO2, OPTIONS | Sets a constant temperature distribution.[a] |
| UPDATE_SURFACE_TENSION | MODE SCO2, OPTIONS | Turns on surface tension scaling. |
| MIXTURE_DENSITY | MODE SCO2, OPTIONS | Sets $CO_2$-brine mixture density.[b] |
| UPWIND_VISCOSITY | MODE SCO2, OPTIONS | Fully upwinds viscosity.[c] |
| PHASE_PARTITIONING | MODE SCO2, OPTIONS | Sets $CO_2$-brine phase partitioning model.[d] |
| NO_H2O_SOURCE_UPDATE_FROM_TRANS | MODE SCO2, OPTIONS | Turns off $H_2O$ source/sink from reaction. |
| TYPE | WELL_MODEL | Sets the well model type.[e] |
| WELLBORE_MODEL | SUBSURFACE | Opens the Wellbore model input block.[f] |
| WELL_MODEL_OUTPUT | SUBSURFACE | Sets well variables to output.[g] |

[a] Temperature distribution is read in through an INITIAL_CONDITION the same way as a thermal model, and it is held constant throughout the simulation.

[b] Options: GARCIA (default is Alendal and Drange, 2001).

[c] Default is a harmonic average.

[d] Options: SPYCHER_SIMPLE. Default is Spycher and Pruess, 2010.

[e] Options: HYDROSTATIC is the only supported option with SCO2 Mode.

[f] This is where the user defines all well properties including well trajectory, geometry, flow rate, and well index relationship. Use multiple WELLBORE_MODEL blocks to define multiple wells.

[g] Options: WELL_LIQ_PRESSURE, WELL_GAS_PRESSURE, WELL_LIQ_Q, WELL_GAS_Q.

Within the SUBSURFACE block, several new options have been added to simulate injection and production wells. The WELLBORE_MODEL block is used to begin defining the properties of a well. Multiple WELLBORE_MODEL blocks can be used to define multiple wells, each with different properties. Each WELLBORE_MODEL block requires a name (e.g., WELLBORE_MODEL well-1, WELLBORE_MODEL well-2). Within a WELLBORE_MODEL block, several sub-blocks are used to build different parts of the well and apply specific constraints. The first sub-block is the WELL_GRID block, which
is used to define the orientation and trajectory of a particular well. The WELL block defines properties of the wellbore itself, like diameter and skin factor. Pressure limitations can be imposed by the FRACTURE_PRESSURE and MINIMUM_PRESSURE keywords. A list of keywords can be found in Table 4, and more information on well model construction can be found in Section 3.3.





### 3.1.1 New Characteristic Curve Options

A new, head-based Van Genuchten characteristic curve option is now available. This model takes the following form:

$$\bar{s}_l = (1 + (\alpha \beta_{gl} h_{gl})^n)^{-m}, \tag{30}$$

$$\bar{s}_l = \frac{s_l - s_{lr}}{1 - s_{lr}}, \tag{31}$$

$$h_{gl} = \frac{P_g - P_l}{\rho_l^* g} \tag{32}$$

where $h_{gl}$ is the gas-liquid capillary head in meters, $\alpha$ is the inverse of the capillary entry head in $m^{-1}$, $\beta_{gl}$ is the interfacial

tension scaling factor, and $\rho_l^*$ is a liquid reference density, which by default is set to 998.32 kg/m$^3$.

This new capillary head-based saturation function is invoked with keyword VG_STOMP. Both the Van Genuchten capillary head function and the default Brooks-Corey capillary pressure function can now be optionally extended into the region below residual liquid saturation by using the UNSATURATED_EXTENSION keyword. Both functions are also now compatible with the scanning path hysteresis model for gas trapping.

**Table 3.** Characteristic Curve Keywords

| Keyword | Description |
|---|---|
| UNSATURATED_EXTENSION | Extends the capillary pressure function below residual saturation. |
| MAX_TRAPPED_GAS_SAT | Sets $s_{gt,max}$ for the hysteresis model. |
| VG_STOMP | Selects the Van Genuchten capillary head model. |
| OVEN_DRIED_CAP_HEAD | Sets head value at $s_l = 0$. * |

* Applies to Van Genuchten capillary head function only, used when extending the capillary head curve below residual saturation.





**Table 4.** Well Model Keywords

| Keyword(s) | Parent Block(s) | Description |
| --- | --- | --- |
| WELL_GRID | WELLBORE_MODEL | Defines well orientation and casing. |
| WELL | WELLBORE_MODEL | Defines well geometric properties. |
| USE_WELL_COUPLER | WELLBORE_MODEL | Sets rate through a WELL_COUPLER.[a] |
| FRACTURE_PRESSURE | WELLBORE_MODEL | Fracture pressure limit.[b] |
| MINIMUM_PRESSURE | WELLBORE_MODEL | Minimum pressure limit.[c] |
| PRINT_WELL_FILE | WELLBORE_MODEL | Prints a .well file |
| WELL_TRAJECTORY | WELL_GRID | Defines well trajectory.[d] |
| SURFACE_ORIGIN | WELL_TRAJECTORY | Center of the top of the well. |
| SEGMENT_LIST | WELL_TRAJECTORY | Reads well segment information.[e] |
| SEGMENT_DXYZ | WELL_TRAJECTORY | Define well by line segments.[f] |
| SEGMENT_RADIUS_TO_HORIZONTAL_X | WELL_TRAJECTORY | Define a kickoff radius.[g] |
| SEGMENT_RADIUS_TO_HORIZONTAL_Y | WELL_TRAJECTORY | Define a kickoff radius.[h] |
| SEGMENT_RADIUS_TO_HORIZONTAL_ANGLE | WELL_TRAJECTORY | Define a kickoff radius.[i] |
| DIAMETER | WELL | Sets wellbore diameter. |
| FRICTION_COEFFICIENT | WELL | Sets axial friction coefficient. |
| SKIN_FACTOR | WELL | Sets well skin factor. |
| WELL_INDEX_MODEL | WELL | Sets well index model.[j] |

[a] A WELL_COUPLER block links FLOW_CONDITIONs and TRANSPORT_CONDITIONs to WELLs in the same manner as INITIAL_CONDITIONs and BOUNDARY_CONDITIONs link FLOW_CONDITIONs and REGIONs.

[b] The well model switches to pressure-controlled if pressures exceed the FRACTURE_PRESSURE anywhere in the well (for injection wells).

[c] The well model switches to pressure-controlled if pressures drop below MINIMUM_PRESSURE anywhere in the well (for extraction wells).

[d] Other options exist (i.e., TOP_OF_HOLE / BOTTOM_OF_HOLE for vertical wells, CASING for cell-by-cell casing specification). Please see the PFLOTRAN Documentation for alternative keywords for defining well trajectory.

[e] This can be written into the input deck or read from a FILE

[f] This marks a change in <dx,dy,dz> from the reference point, starting at SURFACE_ORIGIN and updating with each segment. Each segment must be defined as CASED or SCREENED/PERFORATED/UNCASED. This is used instead of the SEGMENT_LIST keyword. PFLOTRAN will determine segment lengths and centers based off of the model geometry and save a segment list file for future use. Use in combination with multiple SEGMENT_DXYZ keywords and/or SEGMENT_RADIUS_TO_HORIZONTAL_X / Y / ANGLE.

[g] Ends in a horizontal leg in the x-direction.

[h] Ends in a horizontal leg in the y-direction.

[i] Ends in a horizontal leg at an angle between the x- and y- axes.

[j] Options: PEACEMAN_ISO, PEACEMAN_2D, PEACEMAN_3D, SCALE_BY_PERM. Default is PEACEMAN_3D.

## 3.2 Isothermal Injection

Setting up an isothermal model with a $CO_2$ injection requires first invoking the SCO2 flow mode, providing an isothermal temperature in the SUBSURFACE_FLOW OPTIONS block, providing SCO2 Mode-specific initial flow conditions, and setting




an SCO2 Mode-specific injection flow condition. Setting up the grid, specifying material properties/characteristic curves, labeling regions, etc can all be performed using standard PFLOTRAN input deck formatting. An example of some minimum requirements for setting up an isothermal $CO_2$ injection model are shown below.

The following example contains sections of an input deck that can be used to set up a simple $CO_2$ injection. Complete input decks corresponding to more complex benchmark problems can be found in the supplementary material.

SCO2 Mode is invoked in the SUBSURFACE_FLOW block with keywords MODE SCO2. To make the model isothermal, the keyword ISOTHERMAL_TEMPERATURE is used in the OPTIONS block. This model will run at a constant isothermal
temperature of 45 C.

```
SIMULATION
  SIMULATION_TYPE SUBSURFACE
  PROCESS_MODELS
    SUBSURFACE_FLOW flow
      MODE SCO2
      OPTIONS
        ISOTHERMAL_TEMPERATURE 45.d0
      /
    /
  /
END
```

Flow conditions for the isothermal version of SCO2 Mode require three constraints. Constraints should be selected
depending on the phase state of the flow condition. For example, if the model is to be initialized at fully aqueous-saturated conditions, a FLOW_CONDITION must be specified with LIQUID_PRESSURE, CO2_MASS_FRACTION, and SALT_MASS_FRACTION. Likewise, if an initial or boundary condition were to be in the two-phase state, a GAS_PRESSURE, LIQUID_PRESSURE, and SALT_MASS_FRACTION would be required. That is, to achieve a particular phase state for a flow condition, the constraints requested in the flow conditions must match the primary variables associated
with that phase state (Table 1).





```
    SUBSURFACE
    ...

FLOW_CONDITION initial
      TYPE
        LIQUID_PRESSURE HYDROSTATIC
        CO2_MASS_FRACTION DIRICHLET
        SALT_MASS_FRACTION DIRICHLET
405   /
      DATUM 0.d0 0.d0 1.d2
      LIQUID_PRESSURE 5.0d6
      CO2_MASS_FRACTION 0.d0
      SALT_MASS_FRACTION 1.d-2
END
    ...
```

The $CO_2$ injection is defined through a RATE type flow condition, where water mass rate, $CO_2$ mass rate, and salt mass rate
are specified followed by their corresponding units. More information on options for specifying injection/extraction rates can
be found in Section 3.3.

```
    FLOW_CONDITION injection
      TYPE
        RATE MASS_RATE
420   /
      RATE 0.d0 12.5d0 0.d0 kg/s kg/s kg/s
    END
    ...
```





## 3.3 Well Model

Defining a well trajectory in PFLOTRAN can take several forms (see Table 4). The most general form invokes the WELL_TRAJECTORY keyword and allows the user to employ an arbitrary combination well path segments (lines and curves) to construct a well geometry of their choice. Since the well trajectory is defined first by a set of lines and curves in 3 dimensions, using this option ensures that if reservoir grid resolution is refined, the well trajectory will be refined accordingly and that well segments will properly map to reservoir cells. Defining the well trajectory always begins at the surface <x,y,z> location of the wellhead and proceeds downward through a sequence of well intervals.

As an example, constructing the well shown in Figure 2 would require four entries to the WELL_TRAJECTORY block: first, the SURFACE_ORIGIN would be defined (red circle in Figure 2). Then, well interval 1 would be defined by SEGMENT_DXYZ and the corresponding dx, dy, and dz of that interval (green in Figure 2). Next, the well transitions to horizontal in the x-dimension over a particular radius. This is achieved by using the SEGMENT_RADIUS_TO_HORIZONTAL_X keyword and the desired radius $r$ (green in Figure 2). Finally, a horizontal leg extending in the x-direction can be constructed with another SEGMENT_DXYZ entry. Note, this interval definition is independent of the reservoir mesh. PFLOTRAN will split each interval into its corresponding discrete well segment form (i.e., the representation in Figure 1) internally as a function of the mesh. For curved intervals or intervals that do not move strictly in one principal direction, this can result in stair-stepping on structured grids, so users should keep this in mind when building wells. Each well interval is additionally defined as being either CASED or SCREENED; separate well intervals need to be defined accordingly. CASED intervals are disconnected from the reservoir cells through which they pass (i.e., well-reservoir flux is 0); SCREENED intervals are hydraulically connected to their reservoir cells.

On very large unstructured grids, the algorithm for constructing discrete well segments and mapping well segments to reservoir cells can be computationally costly. Therefore, whenever a well is constructed using the WELL_TRAJECTORY keyword, PFLOTRAN outputs a <well-name>_well-segments.dat text file containing all of the necessary information to import the discretized well back into the simulator for future runs. It is recommended that the user read in a well trajectory by invoking the keywords SEGMENT_LIST FILE <well-name>_well-segments.dat when running multiple models on the same reservoir mesh with the same well or wells, to speed up model runs. If the reservoir discretization changes, or if the well trajectories are adjusted, these files must be regenerated.





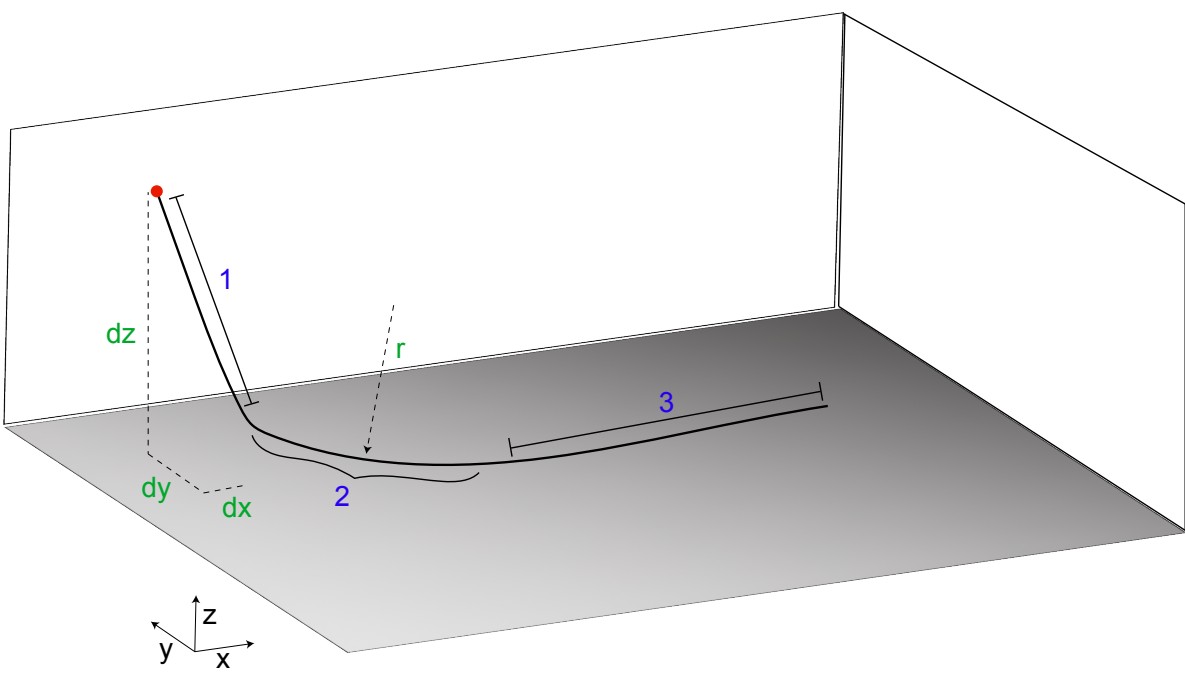

**Figure 2.** Example well trajectory. Red: the location of the wellhead; blue: example ordering of well intervals; green: example input types





The wellbore itself is defined by several parameters. Those include the wellbore diameter, wellbore axial friction coefficient, skin factor, and well index model. The default well index model is the 3D anisotropic Peaceman model (PEACEMAN_3D). To define an upper pressure threshold, beyond which the well is treated as pressure controlled, the keyword FRACTURE_PRESSURE must be invoked followed by a value for fracture pressure. Likewise, a minimum threshold pressure can

be assigned using the keyword MINIMUM_PRESSURE.

It is recommended that the user define well flow rate and composition for SCO2 Mode using a WELL_COUPLER. By invoking the USE_WELL_COUPLER keyword in the WELLBORE_MODEL block, the user can then subsequently define a FLOW_CONDITION to be applied to the well vis-a-vis the WELL_COUPLER block. Just as a BOUNDARY_CONDITION/INITIAL_CONDITION/SOURCE_SINK block links FLOW_CONDITIONs (and TRANS-

PORT_CONDITIONs) to REGIONs, a WELL_COUPLER links FLOW_CONDITIONs (and TRANSPORT_CONDITIONs) to WELLs. A FLOW_CONDITION for a well contains at minimum a TYPE RATE MASS_RATE constraint, where the RATE can be defined by a single set of values or a time-dependent list of values (see PFLOTRAN Documentation for a more detailed description of flow conditions and rate options, including interpolation options).

The RATE keyword expects the same number of entries as there are reservoir degrees of freedom. That means that for

isothermal simulations, RATE expects 3 entries followed by 3 units. The order matters: the first entry is a water source or sink, the second entry is a $CO_2$ source or sink, and the third entry is a salt source or sink. Units are mass per time. Positive numbers indicate injection. Therefore, to inject 1000 kg/s of pure $CO_2$, the RATE would appear as follows:

```
RATE 0.0 1000.0 0.0 kg/s kg/s kg/s
```

If thermal effects are being considered, the RATE keyword expects an additional entry for energy flux. Note that this also

applies to FIXED_TEMPERATURE_GRADIENT, and that the energy source term is added on top of the enthalpy of the fluid being injected. Therefore, it is most common for the energy source/sink to be 0. Units also need to be provided, typically in MW. For example, a modified RATE when running thermal simulations looks as follows:

```
RATE 0.0 1000.0 0.0 0.0 kg/s kg/s kg/s MW
```

By default, injection temperature is set equal to the reservoir temperature at the location of the injection. Injection temper-

ature can optionally be modified by including an additional TEMPERATURE DIRICHLET constraint in the TYPE block of the FLOW_CONDITION. Also, RATEs are treated by default as pure component mass rates. The user can optionally invoke the CO2_MASS_FRACTION DIRICHLET and RELATIVE_HUMIDITY DIRICHLET constraint types. By invoking these, PFLOTRAN will treat water mass source terms as liquid phase mass sources and partition between $CO_2$ and $H_2O$ mass according to the CO2_MASS_FRACTION provided; likewise, a $CO_2$ mass source will be treated as a $CO_2$ phase mass source and

partition between components according to the RELATIVE_HUMIDITY fraction. This could also be achieved by supplying two component mass rates without the need to invoke the CO2_MASS_FRACTION or RELATIVE_HUMIDITY keywords.

Negative RATE values extract mass from the reservoir through the well. Reservoir fluid temperature and composition are always used with extraction wells. Thus, the values provided are always total mass rates. Any values provided to the mass rate





terms are summed, and PFLOTRAN applies that value as the total mass extraction rate applied to the model. Therefore it is
recommended that users supply total mass extraction rates to the first entry of each RATE for simplicity. To extract 1000 kg/s
of total mass in an isothermal simulation, a RATE would look as follows:

```
RATE -1000.0 0.0 0.0 kg/s kg/s kg/s
```

For example, if the reservoir contains only liquid water with dissolved salt and dissolved $CO_2$, applying this rate will extract
water, $CO_2$, and salt components where the sum of the component mass rates equals the total production rate. On the other
hand, if free-phase $CO_2$ and liquid water both coexist in the reservoir cell where the extraction is taking place, individual phase
mass rates are weighted by their mobility ratios (i.e., if gas phase exists but it is immobile, only liquid phase will be extracted,
and vice versa). PFLOTRAN then solves for well bottomhole pressure such that sum of total mass of water and $CO_2$ extracted
from both phases equals the user-specified extraction rate.

When running models with reactive transport, a TRANSPORT_CONDITION will define the constraints on dissolved aque-
ous species that are not $CO_2$. For instance, if the user wishes to inject 1000 kg/s of water with a tracer in an isothermal model,
the FLOW_CONDITION would read as follows:

```
RATE 1000.0 0.0 0.0 kg/s kg/s kg/s
```

where the water component is being injected at 1000 kg/s. A separate TRANSPORT_CONDITION would prescribe the tracer
concentration in the water.

The following example contains sections of an input deck that invoke certain well model options. Complete input decks
corresponding to more complex benchmark problems can be found in the supplementary material.

The well model is first requested by invoking WELL_MODEL in the PROCESS_MODELS block of SIMULATION:

```
SIMULATION
  SIMULATION_TYPE SUBSURFACE
  PROCESS_MODELS
    SUBSURFACE_FLOW flow
      MODE SCO2
    /
    WELL_MODEL well1
      TYPE HYDROSTATIC
    END
  /
END
```



In the WELLBORE_MODEL block in SUBSURFACE, the WELL_GRID and WELL properties must be defined. Optionally, USE_WELL_COUPLER links a well to flow/transport conditions and FRACTURE_PRESSURE sets a maximum pressure threshold for pressure control:

```
SUBSURFACE
      ...

      WELLBORE_MODEL well1
        PRINT_WELL_FILE
WELL_GRID
          WELL_TRAJECTORY
            SURFACE_ORIGIN 50.d0 50.d0 1000.d0
            SEGMENT_DXYZ CASED 0.d0 0.d0 -200.d0
            SEGMENT_DXYZ SCREENED 0.d0 0.d0 -100.d0
530       /
        /

        WELL
          DIAMETER 0.25
FRICTION_COEFFICIENT 1.d0
          WELL_INDEX_MODEL PEACEMAN_3D
          SKIN_FACTOR 0.d0
        /

USE_WELL_COUPLER

        FRACTURE_PRESSURE 15.0d6

      END
...
```

A separate WELL_MODEL_OUTPUT block defines certain well variables to include in output files.

```
WELL_MODEL_OUTPUT
```





```
WELL_LIQ_PRESSURE
       WELL_GAS_PRESSURE
       WELL_LIQ_Q
       WELL_GAS_Q
     /
...
```

Injection/extraction rates and schedules are defined in a FLOW_CONDITION and then coupled to any/all wells through a WELL_COUPLER.

```
FLOW_CONDITION inject_extract
       SYNC_TIMESTEP_WITH_UPDATE
       TYPE
         RATE MASS_RATE
       /
RATE LIST
         TIME_UNITS y
         DATA_UNITS kg/y kg/y kg/y
         0.d0 0.d0 1.d9 0.d0
         2.d0 0.d0 0.d0 0.d0
3.d0 1.d7 0.d0 0.d0
         4.d0 -1.d8 0.d0 0.d0
       /
     END
     ...
     WELL_COUPLER injection-extraction
       FLOW_CONDITION inject_extract
       WELL well1
     END
...
```





## 3.4 Non-isothermal Flow

Running non-isothermal models is the default behavior of SCO2 Mode. Therefore to set up a $CO_2$ model that considers thermal effects, no additional OPTIONS are required in the SUBSURFACE_FLOW block. However, in addition to the three mass degree of freedom constraints, an additional constraint for the energy equation is also required. The fourth primary variable for all phase states is temperature; therefore, a temperature Dirichlet constraint (or energy flux Neumann constraint) is required in each FLOW_CONDITION. An example of some minimum requirements for setting up a thermal $CO_2$ injection model are shown below.

The following example contains sections of an input deck that can be used to set up a simple $CO_2$ injection considering thermal effects. Complete input decks corresponding to more complex benchmark problems can be found in the supplementary material.

SCO2 Mode is invoked in the SUBSURFACE_FLOW block with keywords MODE SCO2. No additional options are required for a thermal model.

```
SIMULATION
  SIMULATION_TYPE SUBSURFACE
  PROCESS_MODELS
    SUBSURFACE_FLOW flow
      MODE SCO2
    /
  /
END
```





In each FLOW_CONDITION, a thermal constraint must be added. For temperature constraints of TYPE DIRICHLET,
the user can either specify a constant temperature or the temperature at a reference datum in conjunction with a geothermal
gradient. Here, the initial condition specifies a geothermal temperature gradient and a temperature at a specified datum. In an
injection well, a constant temperature is specified.

```
     SUBSURFACE
...

     FLOW_CONDITION initial
       TYPE
         LIQUID_PRESSURE HYDROSTATIC
CO2_MASS_FRACTION DIRICHLET
         SALT_MASS_FRACTION DIRICHLET
         TEMPERATURE DIRICHLET
       /
       DATUM 0.d0 0.d0 1.d2
GRADIENT
         TEMPERATURE 0.d0 0.d0 -2.d-2
       /
       LIQUID_PRESSURE 5.0d6
       CO2_MASS_FRACTION 0.d0
SALT_MASS_FRACTION 1.d-2
       TEMPERATURE 45.d0
     END
     ...

FLOW_CONDITION injection_well
       SYNC_TIMESTEP_WITH_UPDATE
       TYPE
         RATE MASS_RATE
         TEMPERATURE DIRICHLET
635    /
       TEMPERATURE 30.d0
       RATE LIST
         TIME_UNITS y
```





```
       DATA_UNITS kg/y kg/y kg/y MW
0.d0  0.d0 5.d8 0.d0 0.d0
       3.d0 1.d8 0.d0 0.d0 0.d0
       10.d0 0.d0 0.d0 0.d0 0.d0
    /
END
...
```

## 3.5  Reactive Transport Coupling

SCO2 Mode has been designed to couple with PFLOTRAN's reactive transport libraries for modeling coupled multiphase flow and $CO_2$ reactivity. Invoking this coupling only requires adding a SUBSURFACE_TRANSPORT block to the PROCESS_MODELS list and including $CO_2$ species in additional input blocks associated with reactive transport.

The following example contains sections of an input deck that can be used to set up reaction coupling with $CO_2$ flow. Complete input decks corresponding to more complex example problems, including problems that involve geochemistry coupling, can be found in the supplementary material.

Here, SCO2 Mode is invoked in the SUBSURFACE_FLOW block, and GIRT Mode is invoked through the SUBSURFACE_TRANSPORT block. No additional options are required for reactive transport coupling, but the 655 FIXED_TEMPERATURE_GRADIENT option and the WELL_MODEL block are included here as an example of possible combinations of process models.

```
SIMULATION
  SIMULATION_TYPE SUBSURFACE
PROCESS_MODELS
    SUBSURFACE_FLOW flow
      MODE SCO2
      OPTIONS
        FIXED_TEMPERATURE_GRADIENT
665    /
    /
    SUBSURFACE_TRANSPORT
     MODE GIRT
    /
WELL_MODEL well1
      TYPE HYDROSTATIC
    END
```





```
    /
    END
```

The CHEMISTRY block and associated constraints must include CO2(aq) in order to properly enable coupling between SCO2 Mode and reactive transport. CO2(g) must also be included as an ACTIVE_GAS_SPECIES. This example includes the mineral calcite in its chemistry block, which is defined in a chemistry database. Specific combinations of minerals and aqueous species will depend on the desired application. The PFLOTRAN Documentation contains more information on setting

up reactive transport models, including a description of the structure of a geochemical database.

```
    SUBSURFACE
    ...
```
```
CHEMISTRY
      PRIMARY_SPECIES
        Al+++
        Ca++
        Fe++
CO2(aq)
        ...
      /
```
```
      SECONDARY_SPECIES
HCO3-
        OH-
        CO3--
        CaCO3(aq)
        ...
700   /
```
```
      ACTIVE_GAS_SPECIES
        GAS_TRANSPORT_IS_UNVETTED
        CO2(g)
...
      /
```





```
        MINERALS
        Calcite
        ...
710   /
      ...
    /
    ...

WELL_COUPLER injection
        FLOW_CONDITION injection
        TRANSPORT_CONDITION background_conc
        WELL well1
    END
...
```





## 4 Benchmarks

Five benchmark test cases were built to demonstrate the capability of SCO2 Mode to simulate multiple coupled processes related to subsurface $CO_2$ injection, brine mixing, multiphase miscible flow with hysteresis, well modeling, and reactivity.

These models were constructed from STOMP-CO2 short course material to directly compare PFLOTRAN's results with those produced by STOMP-CO2.

### 4.1 Radial Flow of Supercritical $CO_2$

This benchmark problem is equivalent to STOMP-CO2 Short Course Example Problem CO2-1. In this problem, originally developed for the GeoSeq Project (Pruess et al., 2002), a 1D radial domain is initialized to a constant pressure of 12 MPa and

a temperature of 45 C. The domain is initially water-saturated; at the beginning of the simulation a constant scCO$_2$ injection rate of 12.5 kg/s is applied at r = 0 m and held constant throughout the simulation. The aquifer is homogeneous and isotropic, and the far edge of the model is placed such that the domain is essentially infinite. Gravity effects are ignored.

This benchmark test was designed to evaluate the ability of simulators to properly model two-phase flow of $CO_2$ and brine under capillary and relative permeability effects; to adequately account for temperature, pressure, and salinity dependent phase

properties like density, viscosity, and $CO_2$ solubility; and to appropriately handle phase behavior like $CO_2$ bubbling and salt precipitation during desiccation. This problem has two variants: first, the $CO_2$ injection occurs in a freshwater aquifer; second, $CO_2$ is injected into a brine aquifer. Injection is simulated using a simple source term in the first grid cell (not a well model). Table 5 summarizes key properties of the model.

**Table 5.** Radial Flow of Supercritical $CO_2$: Model Properties

| Property | Value | Units | Description |
|---|---|---|---|
| $\phi$ | 0.12 | - | porosity |
| $k$ | 1.0E-13 | $m^2$ | permeability |
| $T$ | 45 | C | temperature |
| $\alpha$ | 0.5 | 1/m | inverse entry head |
| $n$ | 1.84162 | - | Van Genuchten n |
| $m$ | 0.457 | - | Van Genuchten m |
| $S_{rl}$ | 0.3 | - | residual liquid saturation |
| $S_{rg}$ | 0.05 | - | residual gas saturation |
| $P_l$ | 12 | MPa | initial liquid pressure |
| $x^l_{CO_2}$ | 0 | kg/kg | initial $CO_2$ mass fraction |
| $m_{NaCl}$ | 0.15 | kg/kg | initial salt mass fraction* |
| $q_{CO_2}$ | 12.5 | kg/s | $CO_2$ mass injection rate |

* Salt mass fraction is zero for the first problem variant.





### 4.1.1 Zero Salinity

In this variant of the Radial Flow of Supercritical $CO_2$ benchmark problem, $CO_2$ is injected into a freshwater aquifer (no dissolved salt). Both PFLOTRAN and STOMP-CO2 simulators were run with the same model setup; plots of a select set of output variables vs radial distance from the wellbore are shown below. These output variables include gas pressure, gas saturation, and dissolved $CO_2$ mass fraction (Figure 3). For this problem, PFLOTRAN and STOMP-CO2 results are nearly indistinguishable, indicating very strong agreement between the two simulators and verifying the implementation in PFLOTRAN.



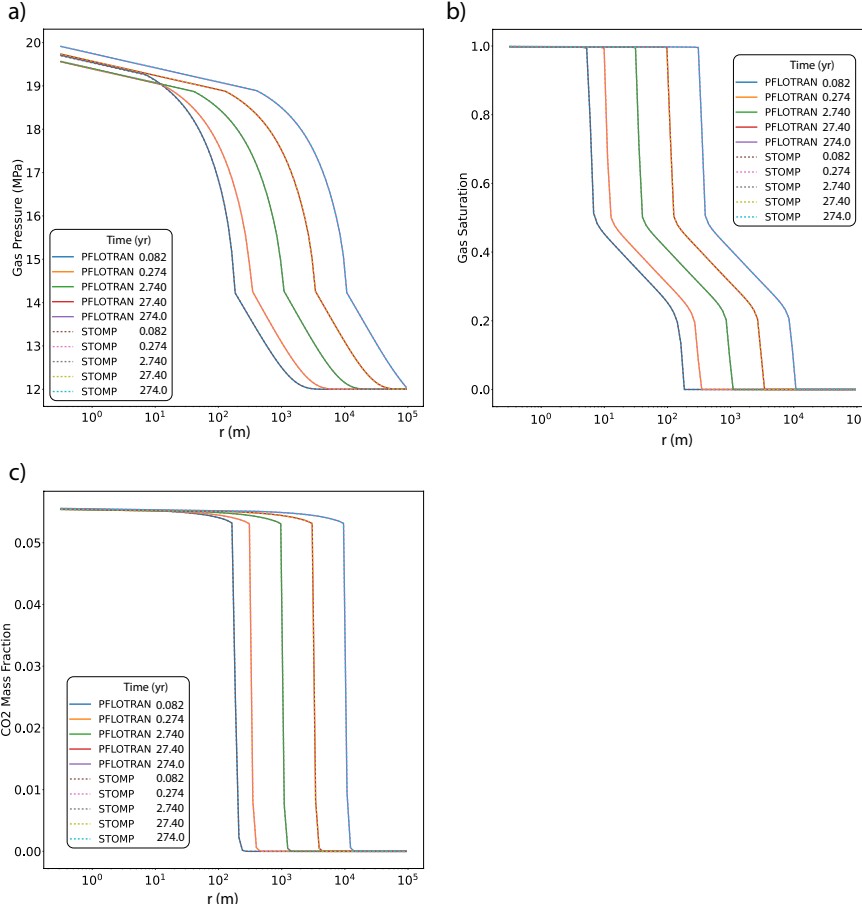

**Figure 3.** Radial Flow of Supercritical $CO_2$ without salt: a) Gas Pressure, b) Gas Saturation, c) $CO_2$ Aqueous Dissolved Mass Fraction.





**4.1.2 With Salinity**

In this variant of the Radial Flow of Supercritical $CO_2$ benchmark problem, $CO_2$ is injected into a brine aquifer characterized by an initial constant dissolved salt mass fraction. Both PFLOTRAN and STOMP-CO2 simulators were run with the same model setup; plots of a select set of output variables vs radial distance from the wellbore are shown below. These output variables include gas pressure, gas saturation, dissolved $CO_2$ mass fraction, and salt precipitate saturation (Figure 4). For this problem,
PFLOTRAN and STOMP-CO2 results are very close for all variables, indicating very strong agreement between the two simulators and verifying the implementation in PFLOTRAN. PFLOTRAN and STOMP-CO2 differ slightly in their reporting of precipitate saturation very close to the wellbore; here, both simulators show a small amount of numerical oscillation. Under strict time step control to small time steps, this numerical oscillation can be eliminated.





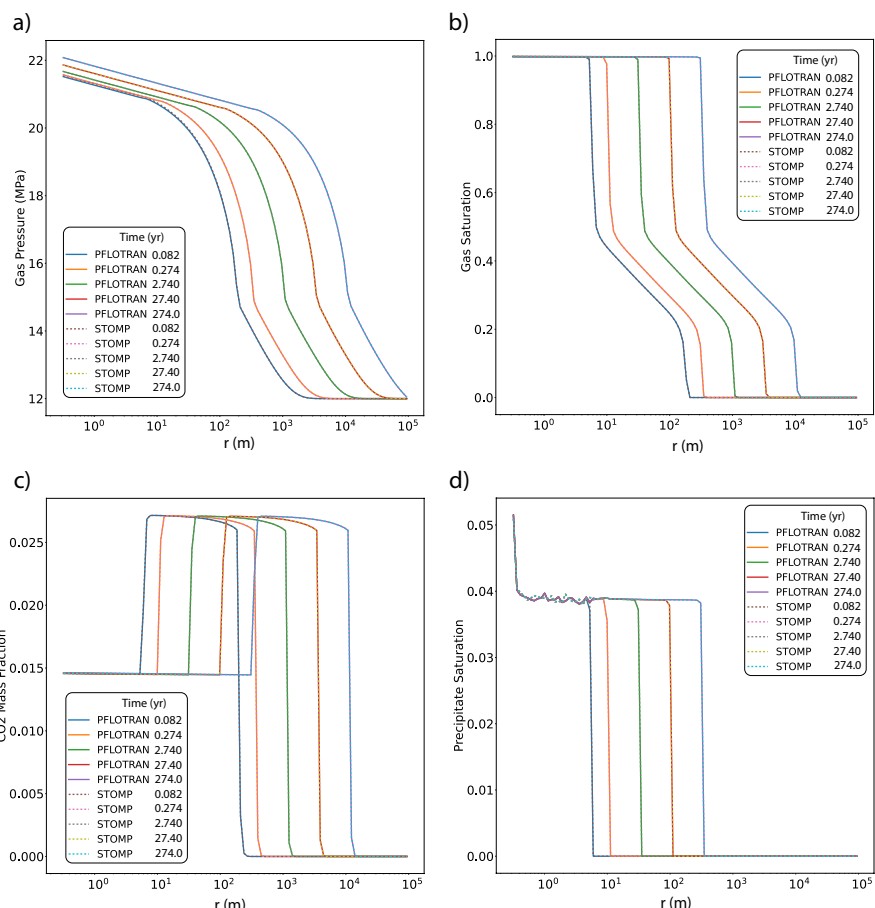

**Figure 4.** Radial Flow of Supercritical $CO_2$ with salt: a) Gas Pressure, b) Gas Saturation, c) $CO_2$ Aqueous Dissolved Mass Fraction, d) Salt Precipitate Saturation.



## 4.2 Discharge of CO$_2$ Along a Fault

This benchmark problem is equivalent to STOMP-CO2 Short Course Example Problem CO2-2. Like the Radial Flow of
Supercritical CO$_2$ benchmark problem, this problem was also introduced as part of the GeoSeq Project (Pruess et al., 2002).
The focus here is on modeling buoyant vertical flow rather than radial flow. Here, a 1D vertical Cartesian domain is initialized
to hydrostatic conditions with pure water (no dissolved salt). This vertical model is meant to emulate loss of stored CO$_2$ into a
freshwater aquifer along a leaky fault. This problem tests the ability of simulators to model displacement of water by buoyant
CO$_2$ migration, dissolution of CO$_2$ at the gas-water interface, hydrostatic water pressure and a Dirichlet top water pressure
boundary, and a bottom gas pressure boundary that drives CO$_2$ flow into the domain.

Pressure at the top of the domain (500 m) is set to 10 MPa, and temperature everywhere is set to 45$^o$C. The model is
isothermal. The model is run in two stages: first, an equilibration step obtains an initial pressure distribution (though the
keyword HYDROSTATIC in PFLOTRAN can also be used to apply a hydrostatic pressure distribution); second, the model is
restarted with a gas phase boundary condition imposed on the bottom boundary and characterized by a gas pressure of 24 MPa.
Table 7 summarizes key properties of the model.

**Table 6.** Buoyant CO$_2$ Flow: Model Properties

| Property | Value | Units | Description |
|---|---|---|---|
| $\phi$ | 0.35 | - | porosity |
| $k$ | 1.0E-13 | $m^2$ | permeability |
| $T$ | 45 | C | temperature |
| $\alpha$ | 0.5 | 1/m | inverse entry head |
| $n$ | 1.84162 | - | Van Genuchten n |
| $m$ | 0.457 | - | Van Genuchten m |
| $S_{rl}$ | 0.3 | - | residual liquid saturation |
| $S_{rg}$ | 0.05 | - | residual gas saturation |
| $P_l$ | 10 | MPa | initial liquid pressure at the top of the model |
| $x^l_{CO_2}$ | 0 | kg/kg | initial CO$_2$ mass fraction |
| $m_{NaCl}$ | 0 | kg/kg | initial salt mass fraction |
| $P_g$ | 24 | MPa | gas pressure applied at the base of the model |

Both PFLOTRAN and STOMP-CO2 simulators were run with the same model setup; plots of a select set of output variables
versus height of the column are shown below. These output variables include gas pressure, gas saturation, aqueous dissolved
CO$_2$ mass fraction, and mass fraction of CO$_2$ in the gas phase (Figure 5). For this problem, PFLOTRAN and STOMP-CO2
results are nearly indistinguishable, indicating very strong agreement between the two simulators and verifying the implemen-
tation in PFLOTRAN.



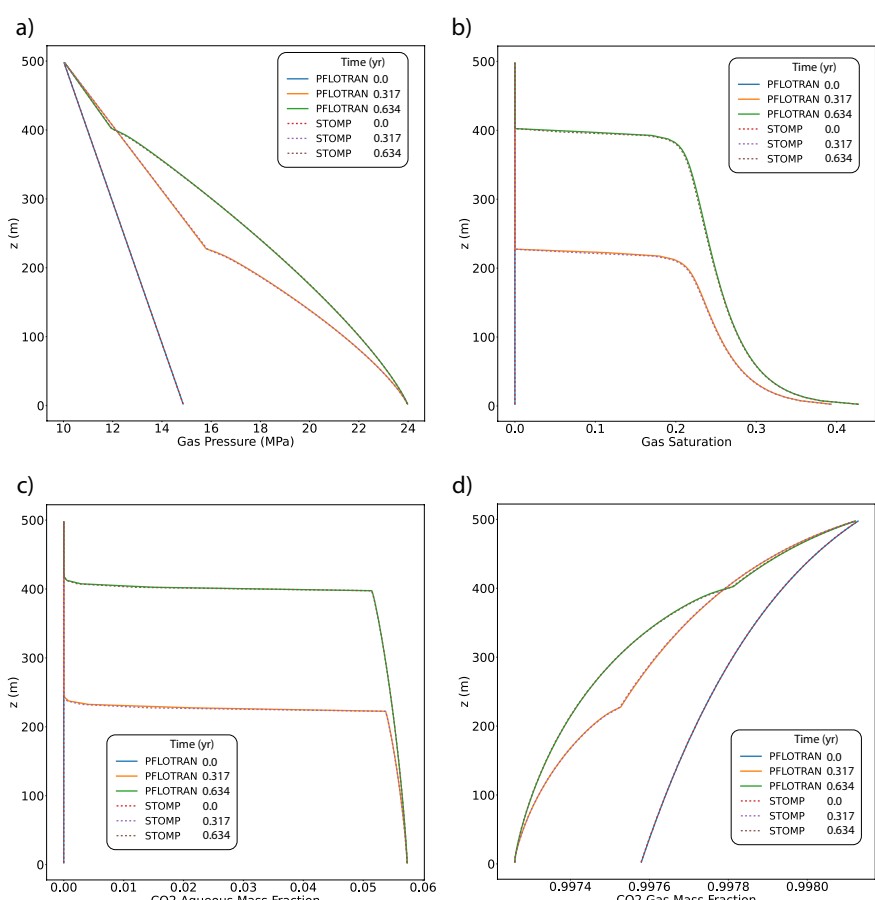

**Figure 5.** CO$_2$ Discharge Along a Fault: a) Gas Pressure, b) Gas Saturation, c) CO$_2$ Aqueous Dissolved Mass Fraction, and d) Free-phase CO$_2$ Gas Mass Fraction.





## 4.3 CO$_2$ Injection into a 2D Layered Formation

This benchmark problem is equivalent to STOMP-CO2 Short Course Example Problem CO2-3. This problem was originally developed for the GeoSeq project (Pruess et al., 2002). Here, a 2D Cartesian layered model was designed to be representative
of the Sleipner Vest field in the Norwegian North Sea. The problem contains alternating sand and shale sequences in a marine environment (i.e., saline pore water).

The problem is isothermal at 37$^o$C, and a source term injects scCO$_2$ into the bottom left corner of a sandy reservoir unit beneath the deepest shale layer. The model is initialized to hydrostatic liquid pressure with a pressure of 11.20525 MPa at the elevation of the injection well, 22 m from the bottom of the domain. No flow boundary conditions are applied on the left as
a symmetry boundary, top as an impermeable shale seal, and bottom as another impermeable seal. The right boundary is a Dirichlet boundary at hydrostatic conditions. The domain spans 6 km horizontally and 184 m vertically, with a series of four 3-m thick horizontal shale units at different depths and spanning the entire length of the domain.

The sand layers and shale layers are characterized by different lithologic properties, adding heterogeneity to the model. They have different permeabilities, porosities, and capillary entry pressures. ScCO$_2$ is injected for a period of 2 years at a single well
location in the model and at a constant rate of 0.1585 kg/s. Table 7 summarizes key properties of the model.

**Table 7.** 2D CO$_2$ Injection into a Layered Formation: Model Properties

| Property | Value | Units | Description |
|---|---|---|---|
| $\phi_{sand}$ | 0.35 | - | sand porosity |
| $k_{sand}$ | 3.0E-12 | $m^2$ | sand permeability |
| $\alpha_{sand}$ | 2.735 | 1/m | sand inverse entry head |
| $\phi_{shale}$ | 0.1025 | - | shale porosity |
| $k_{shale}$ | 1.0E-14 | $m^2$ | shale permeability |
| $\alpha_{sand}$ | 0.158 | 1/m | shale inverse entry head |
| $T$ | 37 | C | temperature |
| $n$ | 1.667 | - | Van Genuchten n |
| $m$ | 0.4 | - | Van Genuchten m |
| $S_{rl}$ | 0.2 | - | residual liquid saturation |
| $S_{rg}$ | 0.05 | - | residual gas saturation |
| $P_l$ | 11.20525 | MPa | initial liquid pressure at the well elevation |
| $x^l_{CO_2}$ | 0 | kg/kg | initial CO$_2$ mass fraction |
| $m_{NaCl}$ | 0.032 | kg/kg | initial salt mass fraction |
| $q_{CO_2}$ | 0.1585 | kg/s | CO$_2$ injection rate at the well |
| $z_{well}$ | 22 | m | elevation of the well |





This problem was used to test the ability of SCO2 Mode to simulate the coupled processes described in the previous benchmarks but in 2D and with lithologic heterogeneity. Simulation results are shown for PFLOTRAN, STOMP-CO2, and published results using TOUGH2 (Pruess et al., 2002) at 30 days, 1 year, and 2 years (Figure 6). ScCO$_2$ initially migrates buoyantly upward from the injection site through the sand reservoir until it reaches a lower permeability shale layer, where it begins to spread

laterally. Over time, due to capillary entry pressure and permeability contrasts, free-phase CO$_2$ accumulates at the interface between layers in the model. Overall, there is good agreement between PFLOTRAN, STOMP-CO2, and TOUGH2, verifying the implementation in PFLOTRAN (Figure 6). PFLOTRAN exhibits slightly more curvature in the the free-phase CO$_2$ plume than STOMP-CO2, which is more plug-like. The PFLOTRAN model more closely matches TOUGH2 in this regard.



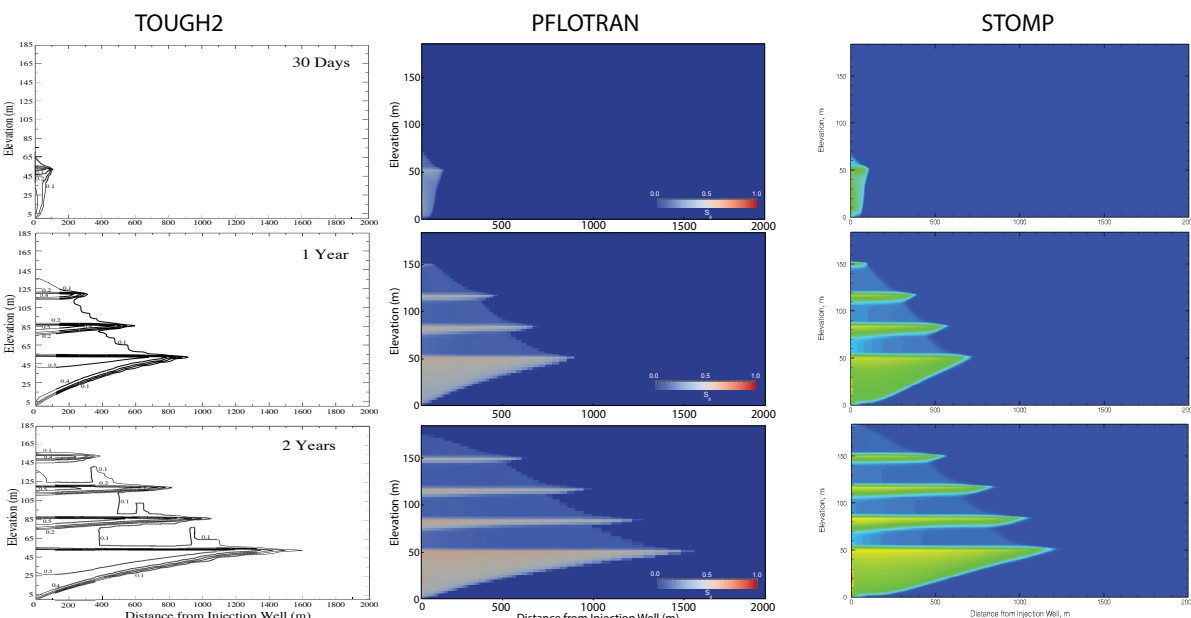

**Figure 6.** TOUGH2, PFLOTRAN, and STOMP-CO2 Comparison. TOUGH2 results adopted from (Pruess et al., 2002), and STOMP-CO2 results adopted from STOMP-CO2 Short Course Problem CO2-3.





### 4.4 2D Cylindrical Injection with a Fully Coupled Well Model

This benchmark problem is equivalent to STOMP-CO2 Short Course Example Problem CO2-4. In addition to the coupled processes verified by the first three benchmark test cases, this problem also tests the ability of SCO2 Mode to model fully coupled wells with pressure control, unsaturated capillary pressure curve extensions, trapped gas hysteresis, permeability anisotropy, fixed temperature gradient, and gravity in a 2D cylindrical domain. Here, a well model is used to inject $scCO_2$ into a layered brine reservoir with heterogeneous permeability and capillary entry pressure.

The model consists of 1 caprock layer at the top of the domain and 24 reservoir layers of varying thickness (Table 8). The original model, developed for STOMP-CO2, uses a mixture of Imperial and SI units; here, all units were converted to SI units. Maximum trapped gas saturation for trapped gas hysteresis calculations is set at 0.2 for all layers in the domain. The model is initialized to hydrostatic pressure and a geothermal gradient of 20$^o$C/km , with a pressure of 12.34 MPa and temperature of 43.75 $^o$C at an elevation of -1040 m. Initial salt mass fraction in the brine is set to 0.0475 kg/kg. The fracture pressure of the 805 reservoir is set to 16.4 MPa; if pressures in the wellbore exceed this pressure, the well becomes rate-controlled. The wellbore diameter is set to 0.2286 m with no well skin. The well injects $CO_2$ at a constant rate of 1.0 MMT/year for the first 2 years. Then the injection well switches off and the model continues to run for a total simulation time of 10 years. The cylindrical grid increases in the radial direction from a cell radius of 5.2 m in the innermost cell to 551.2 m at the far edge for a total of 20 cells in the radial direction. In the vertical dimension, cell thickness varies from 1.2 m to 24.99 m in thickness over a total of 810 25 cells.

PFLOTRAN and STOMP-CO2 results are nearly identical (Figure 7). They show slight differences in the radial extent of the $CO_2$ plume in a few of the cells at the outer edge of the plume. Other than this, the magnitude and distribution of free phase $CO_2$ saturation match between the two simulators, verifying the implementation of the well model with pressure control, trapped gas hysteresis, and capillary pressure extensions in PFLOTRAN.





**Table 8.** 2D Cylindrical Injection: Model Properties

| Material Name | $\phi$ | $k(m^2)$ | Anisotropy$^*$ | Compressibility $(Pa^{-1})^{**}$ | $P_{c,entry}$ (kPa) | $\lambda$ | $S_{rl}$ |
|---|---|---|---|---|---|---|---|
| Caprock | 0.07 | 5.63E-20 | 1.0 | 1.08E-13 | 98 | 0.8311 | 0.0597 |
| Reservoir24 | 0.13 | 1.91E-14 | 1.0 | 5.38E-14 | 11.76 | 0.8311 | 0.0597 |
| Reservoir23 | 0.18 | 1.73E-13 | 1.0 | 5.38E-14 | 4.90 | 0.6215 | 0.0810 |
| Reservoir22 | 0.09 | 1.75E-15 | 0.1 | 5.38E-14 | 35.28 | 0.8311 | 0.0597 |
| Reservoir21 | 0.10 | 4.66E-15 | 1.0 | 5.38E-14 | 19.60 | 0.8311 | 0.0597 |
| Reservoir20 | 0.08 | 7.03E-16 | 0.1 | 5.38E-14 | 49.00 | 0.8311 | 0.0597 |
| Reservoir19 | 0.11 | 1.53E-14 | 1.0 | 5.38E-14 | 12.74 | 0.8311 | 0.0597 |
| Reservoir18 | 0.09 | 2.71E-16 | 0.1 | 5.38E-14 | 67.62 | 0.8311 | 0.0597 |
| Reservoir17 | 0.11 | 7.17E-15 | 1.0 | 5.38E-14 | 16.66 | 0.8311 | 0.0597 |
| Reservoir16 | 0.08 | 3.75E-16 | 0.1 | 5.38E-14 | 60.76 | 0.8311 | 0.0597 |
| Reservoir15 | 0.10 | 5.02E-15 | 1.0 | 5.38E-14 | 18.62 | 0.8311 | 0.0597 |
| Reservoir14 | 0.08 | 1.32E-15 | 1.0 | 5.38E-14 | 30.38 | 0.8311 | 0.0597 |
| Reservoir13 | 0.10 | 5.26E-15 | 1.0 | 5.38E-14 | 18.62 | 0.8311 | 0.0597 |
| Reservoir12 | 0.12 | 1.57E-14 | 1.0 | 5.38E-14 | 11.76 | 0.8311 | 0.0597 |
| Reservoir11 | 0.19 | 2.07E-13 | 1.0 | 5.38E-14 | 3.92 | 1.1663 | 0.0708 |
| Reservoir10 | 0.12 | 1.37E-14 | 1.0 | 5.38E-14 | 12.74 | 0.8311 | 0.0597 |
| Reservoir9 | 0.12 | 1.37E-14 | 1.0 | 5.38E-14 | 12.74 | 0.8311 | 0.0597 |
| Reservoir8 | 0.13 | 2.07E-14 | 1.0 | 5.38E-14 | 10.78 | 0.8311 | 0.0597 |
| Reservoir7 | 0.11 | 6.42E-15 | 1.0 | 5.38E-14 | 16.66 | 0.8311 | 0.0597 |
| Reservoir6 | 0.09 | 2.23E-15 | 1.0 | 5.38E-14 | 25.48 | 0.8311 | 0.0597 |
| Reservoir5 | 0.12 | 1.78E-16 | 1.0 | 5.38E-14 | 63.70 | 0.8311 | 0.0597 |
| Reservoir4 | 0.12 | 4.69E-17 | 1.0 | 5.38E-14 | 104.86 | 0.8311 | 0.0597 |
| Reservoir3 | 0.15 | 1.23E-14 | 1.0 | 5.38E-14 | 13.72 | 0.8311 | 0.0597 |
| Reservoir2 | 0.11 | 2.83E-15 | 1.0 | 5.38E-14 | 22.54 | 0.8311 | 0.0597 |
| Reservoir1 | 0.11 | 2.83E-15 | 1.0 | 5.38E-14 | 22.54 | 0.8311 | 0.0597 |

$^*$ Vertical : Horizontal permeability anisotropy ratio

$^{**}$ linear porosity compressibility





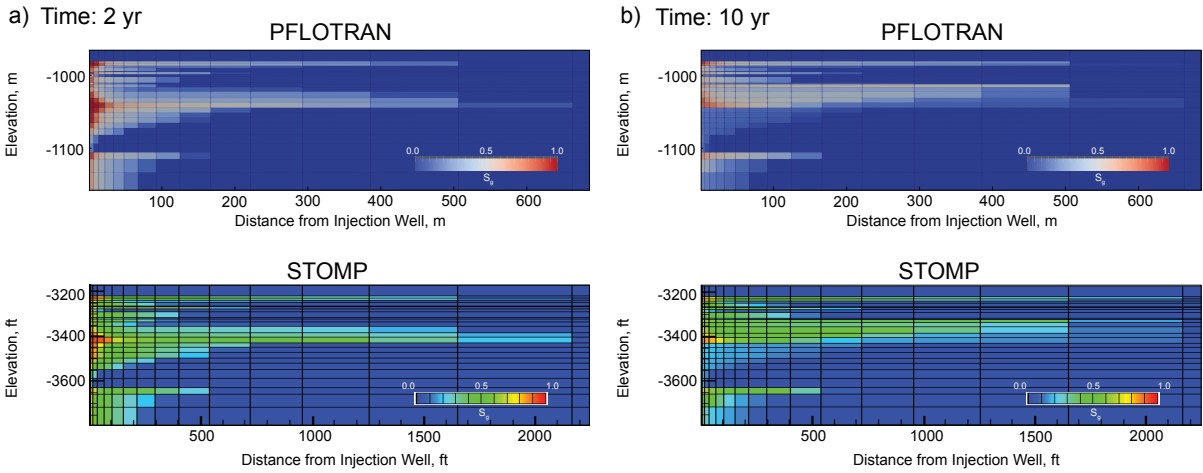

**Figure 7.** Well Model with Gas Trapping Hysteresis: a) 2 years and b) 10 years. STOMP-CO2 results adopted from STOMP-CO2 Short Course Problem CO2-4.





## 4.5  Mineral Trapping in Basalts

This benchmark problem is equivalent to STOMP-CO2 Short Course Example Problem CO2-6. This problem simulates injection of $CO_2$ into a reactive basalt reservoir by coupling SCO2 Mode with the GIRT reactive transport mode in PFLOTRAN. The physical and geochemical properties in this scenario are designed to emulate those of the Grande Ronde continental flood basalts of the Columbia Plateau province in southeastern Washington state. In this system, high permeability basalt flow tops have been targeted as potential reactive reservoirs for $CO_2$ storage; these flow tops are sandwiched between low permeability flow interiors which serve as confining layers and create a stacked reservoir. The Wallula Basalt Pilot Project, which injected roughly 1,000 MT of $CO_2$ into one such reservoir system in 2013, demonstrated the ability of highly reactive basalts to rapidly mineralize $CO_2$ in the subsurface (McGrail et al., 2017).

Here, a 2D radial injection of $CO_2$ into a single flow top reservoir unit is modeled. No flow boundaries are applied at the top and bottom to simulate a flow top confined between two impermeable flow interiors. The model is initialized with a hydrostatic brine pressure distribution and temperature following a geothermal gradient of 26.8 $^oC$/km, with a brine pressure of 10 MPa and temperature of 39.83 $^oC$ at a datum of -1087.53 m elevation. Salt mass fraction is initially set to 0.01 kg/kg everywhere. $CO_2$ is injected along the entire span of the reservoir unit at r = 0 m at a total mass injection rate of 0.827 kg/s for a total of 1,000 Mt of $CO_2$ injected after 14 days. At the end of 14 days, the $CO_2$ source is shut off and the model is run to a total simulation time of 10 years.

PFLOTRAN and STOMP-CO2 results show good agreement. The pH distribution in the reservoir is controlled by the movement of dissolved $CO_2$ through the system (Figure 8); pH profiles after 10 years show similar distributions and magnitudes between STOMP-CO2 and PFLOTRAN. STOMP-CO2 shows a bit more spread in the pH plume than PFLOTRAN, and the pH decrease reaches slightly further in the radial direction. This is consistent with results obtained from previous benchmarks, but it also is likely affected by stronger mineralization reported by PFLOTRAN (Figure 9). The differences could be due to several factors: first, there are likely differences in PFLOTRAN's GIRT mode implementation of the reactive transport equations as compared to STOMP. Second, just like in previous benchmarks, STOMP-CO2 used a combination of imperial, field, and SI units. These values (depths, pressures, temperatures, etc) were converted to SI units for use in PFLOTRAN. Finally, STOMP-CO2 allows the user to specify a hydrostatic gradient to initialize the model, while PFLOTRAN computes hydrostatic pressure internally relative to a datum. The pressure solution will evolve in response to boundary conditions as well as the evolution of $CO_2$ and salt mass distributions; final pressures could be sensitive to initialization and the far-field boundary pressure.

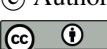


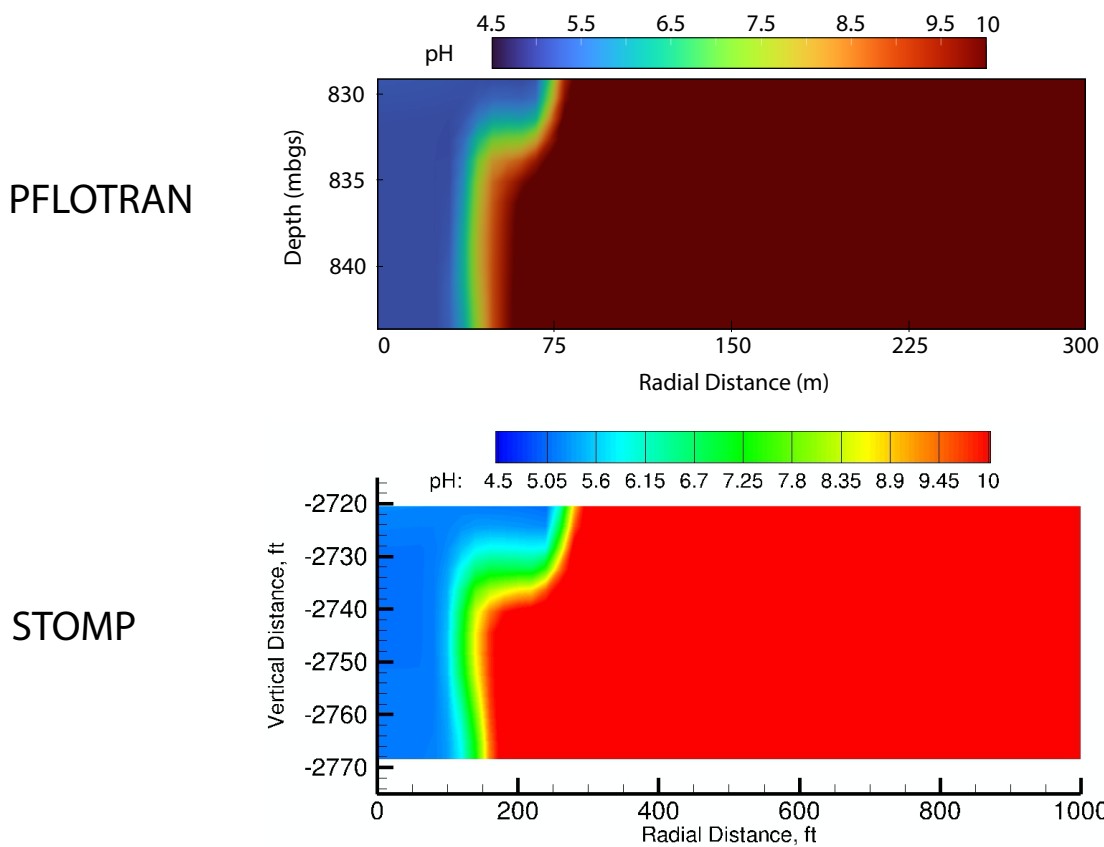

**Figure 8.** pH after 10 years: PFLOTRAN and STOMP-CO2. STOMP-CO2 results adopted from STOMP-CO2 Short Course Problem CO2-6.





There is good agreement between PFLOTRAN and STOMP-CO2 with respect to the evolution of primary and secondary mineralogy in the system (Figure 9). Both simulations agree in terms of order of magnitude of dissolution and precipitation, as well as mineral order of importance. For both models, glass was the most dominant primary mineral to dissolve, followed by

clinopyroxene, plagioclase, and magnetite. Likewise, dolomite was the most dominant secondary carbonate mineral formed, followed by calcite, beidellite-Ca, beidellite-Mg, and then the others. In addition to the differences noted above and in previous benchmark tests, differences in mineral volumes could also have to do with kinetic reaction rates in STOMP-CO2 being supplied at different reference temperatures; reaction rates were adjusted to a common reference temperature of $25^{o}C$ for PFLOTRAN.

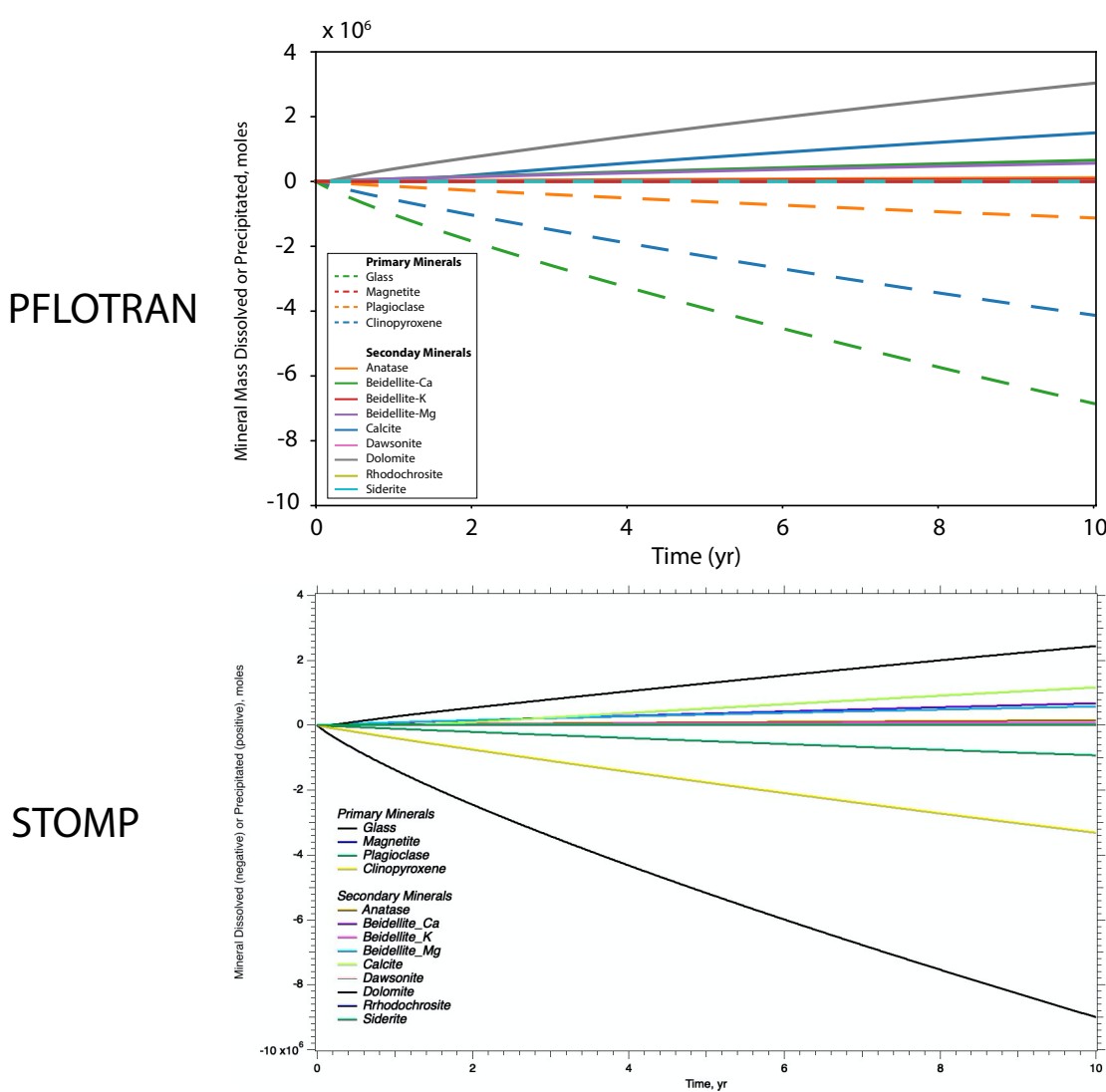

**Figure 9.** Minerals: PFLOTRAN and STOMP-CO2. Negative indicates mineral dissolution, positive indicates mineral precipitation. STOMP-CO2 results adopted from STOMP-CO2 Short Course Problem CO2-6.





The global $CO_2$ mass balance tells a similar story (Figure 10). Both PFLOTRAN and STOMP-CO2 report roughly the same total amount of dissolved $CO_2$ in the whole model, but they differ slightly in the amount of free-phase $CO_2$. This is likely attributable to PFLOTRAN reporting slightly higher amounts of carbonate mineralization in comparison to STOMP-CO2, particularly dolomite mineralization. Mineralization could be retarding the movement of $CO_2$ through the system, which is also expressed in the pH plume. Total free $CO_2$ mass (the sum of dissolved and free gas mass) is consistent between both simulators

at the end of the injection period, and by 10 years PFLOTRAN reports slightly lower total free $CO_2$ mass; the difference is attributable to differences in total mineralization (negligible amounts of $CO_2$ leave the far-field boundary). Overall, good agreement between the two simulators verifies the implementation of $CO_2$ flow coupled to mineral dissolution/precipitation and aqueous reaction networks in PFLOTRAN.





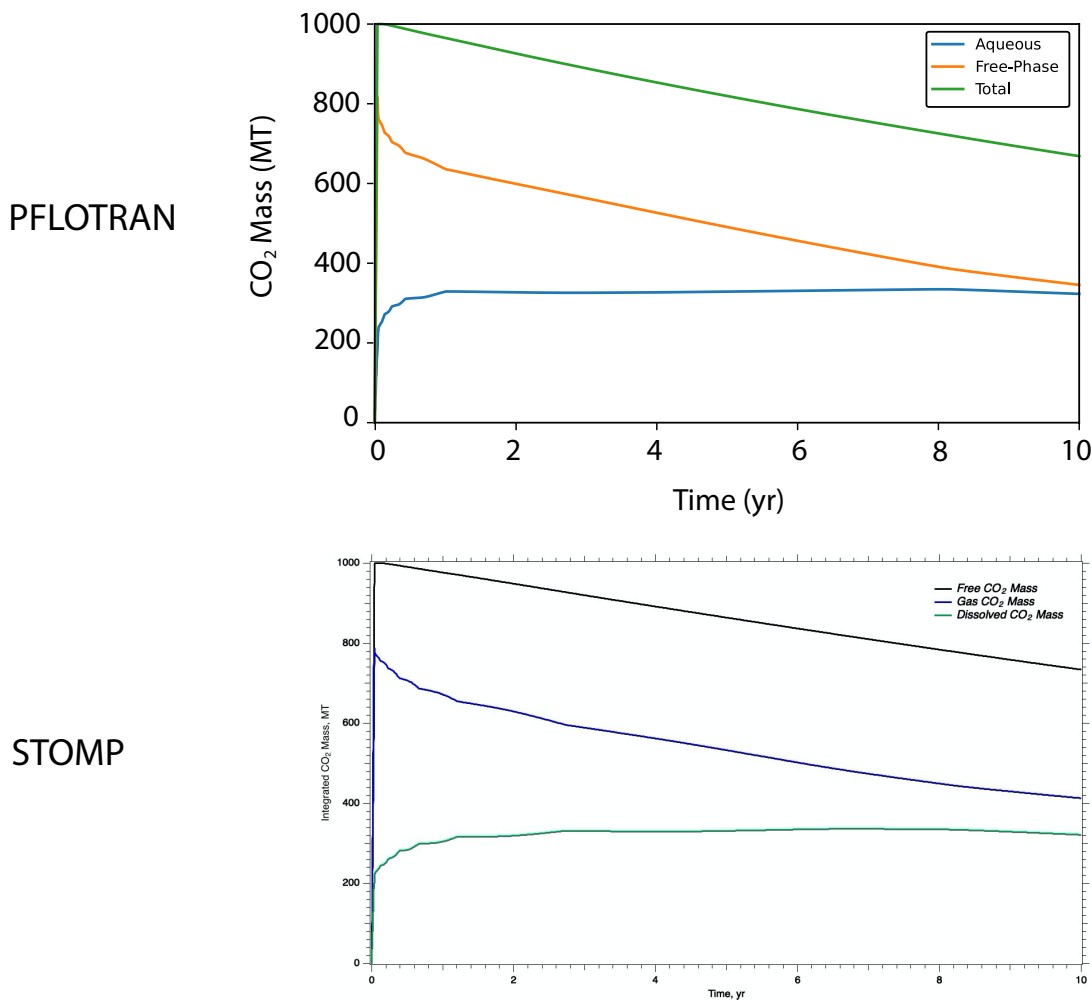

**Figure 10.** Total free (non-mineralized) $CO_2$ mass, aqueous dissolved $CO_2$ mass, and free phase $CO_2$ mass. Total free $CO_2$ mass decreases over time due to mineralization. STOMP-CO2 results adopted from STOMP-CO2 Short Course Problem CO2-6.





## 5  Critical Mineral Extraction

As shown previously, injecting $CO_2$ into subsurface reservoirs naturally acidifies the formation water. In highly reactive basalt rocks, this can dissolve a host rock rich in cations that then react with dissolved $CO_2$ to form secondary carbonate minerals. However, in addition to promoting carbonate reactions in basalts, acidic solutions can also promote leaching of critical minerals from ore deposits; in situ leach mining is an established technique to recover metals from ores in situ such as uranium, gold, copper, and others (Bartlett, 2013). Here we demonstrate how $CO_2$ could be used to promote leaching and in situ production

of a copper ore deposit. This process has the dual benefit of storing $CO_2$ and producing valuable critical minerals.

In this example, a horizontal well is drilled through a hypothetical copper ore deposit. The domain measures 750 m in both lateral directions and 120 m vertically. The ore itself spans the domain laterally and 80 m vertically. It is situated between two impermeable sealing layers. The injection well extends down to the bottom of the ore deposit, where it then kicks off to horizontal for 400 m. The last 300 m of the lateral length of the well are screened; the rest of the well is cased. The

initial mineral composition of the deposit is based on Hammond et al. (2014) and Lichtner (1998) and is shown in Table 9. The reservoir conditions are such that $CO_2$ is a supercritical fluid: formation temperature is $40^oC$ at the top of the domain and increases by $28^oC$/km with depth. Liquid pressure at the top of the domain is 8.3 MPa, and the formation brine has an initial salt mass fraction of 0.01 kg/kg. The well injects $CO_2$ and extracts formation fluids over three cycles for a total copper production period of 20 years. The well first injects 1 kg/s of $CO_2$ for one year, then shuts off for 3 months to allow $CO_2$ to

migrate and react with water to dissolve the rock minerals; the well then extracts formation fluids for one year at a rate of 1 kg/s. The process is repeated twice more, and then the production rate is held constant until Year 20.





**Table 9.** Minerals Considered in the Copper Leaching Problem

| Mineral | Initial Volume Fraction | Rate Constant (mol/cm$^2$-sec) | Activation Energy (kJ/mol) | H$^+$ Prefactor |
|---|---|---|---|---|
| Chrysocolla | 5.0E-3 | 1.0E-10 | - | 0.39 |
| Goethite | 2.5E-2 | 1.0E-11 | - | - |
| Kaolinite | 5.0E-2 | 1.0E-13 | - | - |
| Muscovite | 5.0E-2 | 1.0E-13 | - | - |
| Quartz | 8.7E-1 | 1.0E-14 | - | - |
| Alunite | 0.0 | 1.0E-11 | - | - |
| Aragonite | 0.0 | 5.01E-9 | 14.1 | 0.9 |
| Azurite | 0.0 | 1.55E-10 | 14.1 | 0.9 |
| Calcite | 0.0 | 1.55E-10 | 14.1 | 0.9 |
| Dawsonite | 0.0 | 1.55E-10 | 14.1 | 0.9 |
| Gypsum | 0.0 | 1.0E-10 | - | - |
| Jarosite | 0.0 | 1.0E-11 | - | - |
| Jurbanite | 0.0 | 1.0E-11 | - | - |
| Malachite | 0.0 | 1.55E-10 | 14.1 | 0.9 |
| SiO2(am) | 0.0 | 1.0E-11 | - | - |

After 5.5 years, three cycles of $CO_2$ injections have injected a total of $9.46 \times 10^5$ MT of $CO_2$ through the well. A buoyant free-phase $CO_2$ plume migrates upward through the reservoir above the well; since the $CO_2$ is also dissolved in the pore water, the associated change in pH of the pore water extends beyond the free-phase $CO_2$ plume, which results in a zone of enhanced copper dissolution (Figure 11). In this model, the minimum pH induced by $CO_2$ injection is 4.3. The original formulation of the copper leaching problem from Hammond et al. (2014) injects an aqueous solution with a strong acid of pH 1 into an ore deposit to induce copper leaching. For comparison, we modified our example problem to inject an aqueous solution with the acid used in Hammond et al. (2014) instead of $CO_2$. As expected, a very strong acid initially results in greater amounts of copper extracted from the deposit (Figure 12). But, interestingly, at later time copper extraction rates decay slower with $CO_2$ injection than with acid injection. This could be due to the differences in mobility between $CO_2$ and water and suggests that an optimal $CO_2$ injection strategy could be achieved that balances the timing of injection and production with the rates of free-phase $CO_2$ migration, copper leaching, and copper carbonate mineralization.



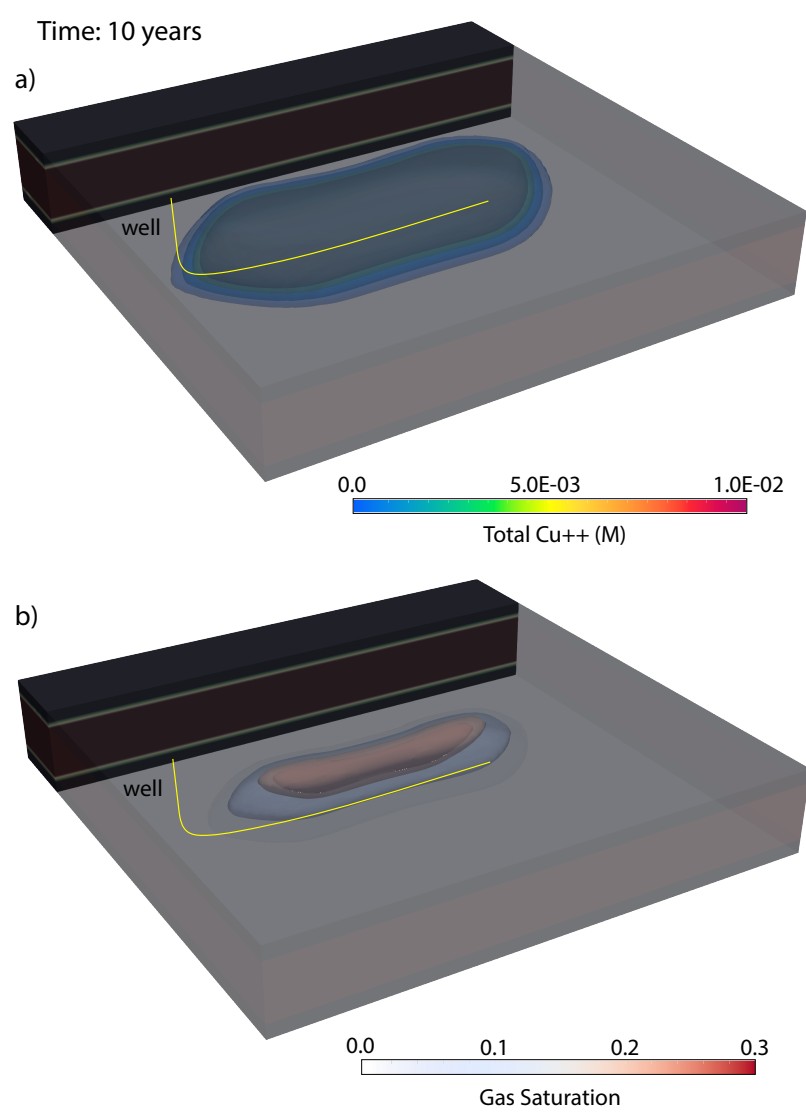

**Figure 11.** A snapshot of a) dissolved copper and b) free-phase $CO_2$ after 10 years.







**Figure 12.** Total copper mass extracted from the well over 10 years, comparing $CO_2$ as the acidification agent to a strong acid (pH=1).







**Figure 13.** Total mineral mass precipitated (positive) or dissolved (negative). Dashed lines indicate primary minerals.

In addition to differences in system pH between a $CO_2$ injection and an acid injection, $CO_2$ can also react with copper cations in solution to form copper carbonates. In this model, the dominant copper carbonate mineral that forms is malachite, while the primary mineral that dissolves to release copper is chrysocolla (Figure 13). Interestingly, silica formation accounts for a significant fraction of the total mass of minerals formed, indicating that silica formation rates could be affecting overall mineral dissolution/precipitation and thus total copper extraction potential.



## 6  Conclusions

Recent developments in the open source multiphase flow and reactive transport simulator PFLOTRAN now allow for modeling
the flow of scCO$_2$ and brine with optional fully implicit coupling to a hydrostatic well model, fully implicit thermal coupling,
and sequential coupling to PFLOTRAN's well-established reactive transport capabilities. Many new features are now available,
including flexible well trajectories, capillary pressure function extensions below residual saturation, and scanning path hystere-
sis for gas trapping. These new features are verified against the STOMP-CO2 simulator for five benchmark test cases, which
show excellent agreement between the two simulators. Finally, a new example problem involving critical mineral extraction is
presented, demonstrating how CO$_2$ can be used as a working fluid for promoting copper leaching from an ore deposit. Critical
mineral extraction using CO$_2$ as the working fluid would take advantage of CO$_2$-induced acidification to leach copper into the
pore water, but the process should be optimized to avoid significant mineralization of secondary copper carbonate minerals.
This example demonstrates how CO$_2$ could be used instead of strong acids to extract critical minerals using a potentially more
environmentally friendly working fluid while simultaneously storing CO$_2$.

*Code and data availability.*  The current version of PFLOTRAN is available from the project website www.pflotran.org under the LGPL3
licence. The exact version of the model used to produce the results used in this paper is archived on Zenodo under DOI 10.5281/zen-
odo.14969296, as are all input data (Nole et al., 2025).





## Appendix A: Verification Tests

A suite of verification tests were constructed to test individual functions, capabilities, or processes in isolation from larger, coupled, system-scale phenomena. These small, 0- and 1-dimensional test cases are designed to evaluate how well the new components developed in SCO2 Mode match their counterparts in STOMP-CO2.

### A1  0D Test Cases

A set of single-cell tests cases was developed to confirm that SCO2 Mode represents the relevant $CO_2$ equations of state, constitutive relationships, initial/boundary conditions, phase behavior, and flow/reaction coupling in a manner that is consistent with STOMP-CO2.

### A1.1  $CO_2$ Injection into a Brine Reservoir

In this test, a single grid cell is initialized at a brine pressure of 12 MPa, initial $CO_2$ mass fraction of 0.0 kg/kg, and initial salt mass fraction of 0.15 kg/kg. $CO_2$ is injected into the cell at a rate of 12.5 kg/s. Capillary pressure is described by a Van Genuchten function with a capillary entry head of 2 m, $n$ equal to 1.84162, liquid residual saturation of 0.3, and gas residual saturation of 0.05. A liquid phase boundary condition on the east face of the cell is set to 12 MPa, a salt mass fraction of 0.22 kg/kg, and no dissolved $CO_2$. An isothermal temperature of 20 $^o$C is set in the cell. PFLOTRAN and STOMP-CO2 report identical results, verifying the implementation of $CO_2$-brine phase partitioning, capillary pressure, salt precipitation models, and liquid phase boundary conditions in PFLOTRAN (Figure A1).

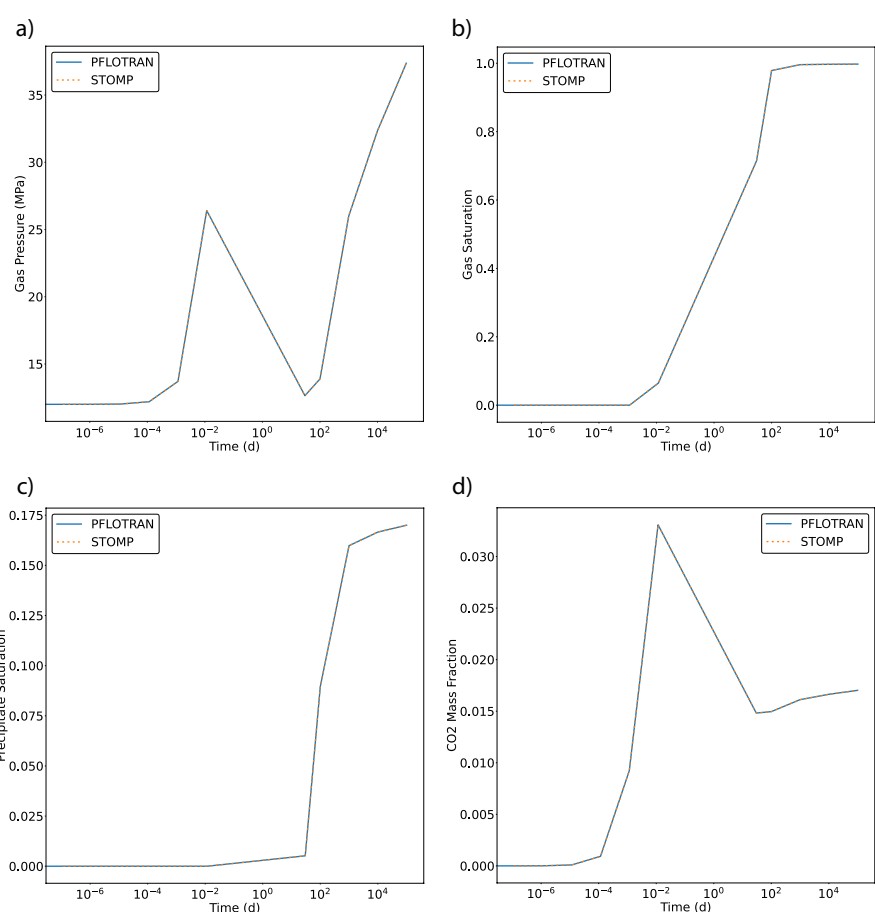

**Figure A1.** Single-cell $CO_2$ Injection: a) Gas Pressure, b) Gas Saturation, c) Precipitate Saturation, and d) Dissolved $CO_2$ Mass Fraction.



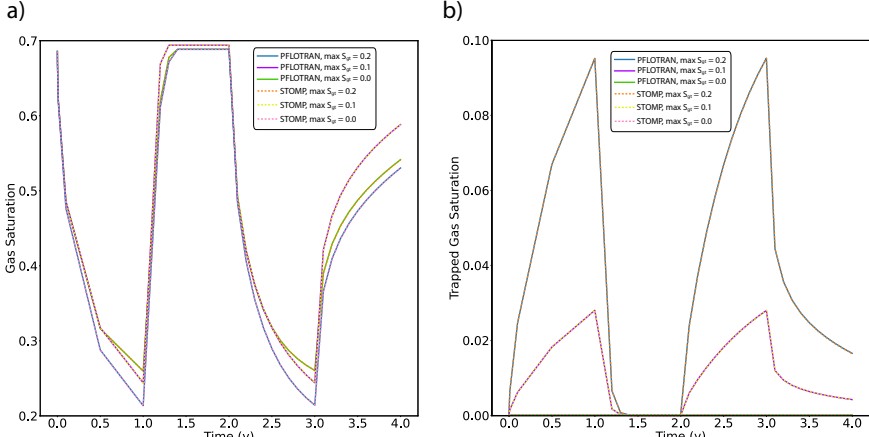

**Figure A2.** Trapped Gas Hysteresis: a) Gas Saturation, b) Trapped Gas Saturation

### A1.2 Trapped Gas Hysteresis

This single-cell test was designed to verify PFLOTRAN's implementation of trapped gas hysteresis over several drainage and imbibition cycles. In this test, a single grid cell is initialized with a brine pressure of 12 MPa, a gas phase pressure of 14 MPa, and no dissolved salt. Using a Van Genuchten capillary pressure function with an entry head of 2 m, Van Genuchten $n$ value of 1.84162, and liquid residual saturation of 0.3, a capillary pressure of 2 MPa is equivalent to an initial gas saturation of 0.6857. In this model, trapped gas hysteresis is enabled through the capillary pressure curve's optional input field for maximum

trapped gas saturation. The model is isothermal at 20 $^oC$, and boundary conditions are applied on the west and east faces of the cell. Both boundary conditions apply a liquid pressure of 12 MPa with no dissolved $CO_2$ or dissolved salt. Initially, the boundary conditions drive water to imbibe into the domain. After 1 year, $CO_2$ is injected into the cell at a rate of 1.0 kg/s, causing water to drain back out of the cell. After 2 years, the gas source is shut off, causing imbibition again. Finally, after 3 years $CO_2$ is injected at a slower rate of 0.1 kg/s until the end of the simulation at 4 years. Three simulations were run with

different values for maximum trapped gas saturation. PFLOTRAN and STOMP-CO2 report identical results for all simulations, verifying PFLOTRAN's implementation of trapped gas hysteresis.





### A1.3 CO₂ Injection into a Reactive Brine Reservoir

This single-cell test contains a batch reactor in which a $CO_2$ injection drives dissolution of primary minerals and subsequent carbonate mineralization. Mineral chemistry is similar to the Mineral Trapping in Basalts benchmark problem chemistry.

The domain is initialized to a liquid pressure of 1 MPa, salt mass fraction of 0.01 kg/kg, and an isothermal temperature of $24^o$C. $CO_2$ is injected at a constant rate of $1.0x10^{-5}$ kg/s for 14 days; the simulation is run for a total of 10 years. pH initially drops in the cell to around a pH of 4; PFLOTRAN and STOMP-CO2 report nearly the same pH drop. The pH then rebounds to close to pH 5 by the end of the model. PFLOTRAN and STOMP-CO2 report nearly the same pH. Primary minerals clinopyroxene and plagioclase dissolve in this model; PFLOTRAN and STOMP-CO2 report very similar dissolution of these

primary minerals over the course of the 10 year simulation. PFLOTRAN and STOMP-CO2 report very similar secondary mineral mass precipitation; some differences at the point of mineral formation could be due to different timestepping in the reactive transport solve or slightly different treatment of mineral nucleation between the two reactive transport solvers. STOMP-CO2 reports nonzero masses of the least abundant secondary minerals (Dawsonite, Rhodochrosite, and Siderate), whereas PFLOTRAN reports those as zero. These mineral masses are 15 orders of magnitude smaller than the dominant secondary

minerals, so they are negligible. This test problem demonstrates an implementation of flow and reactive transport coupling in PFLOTRAN that is consistent with STOMP-CO2.



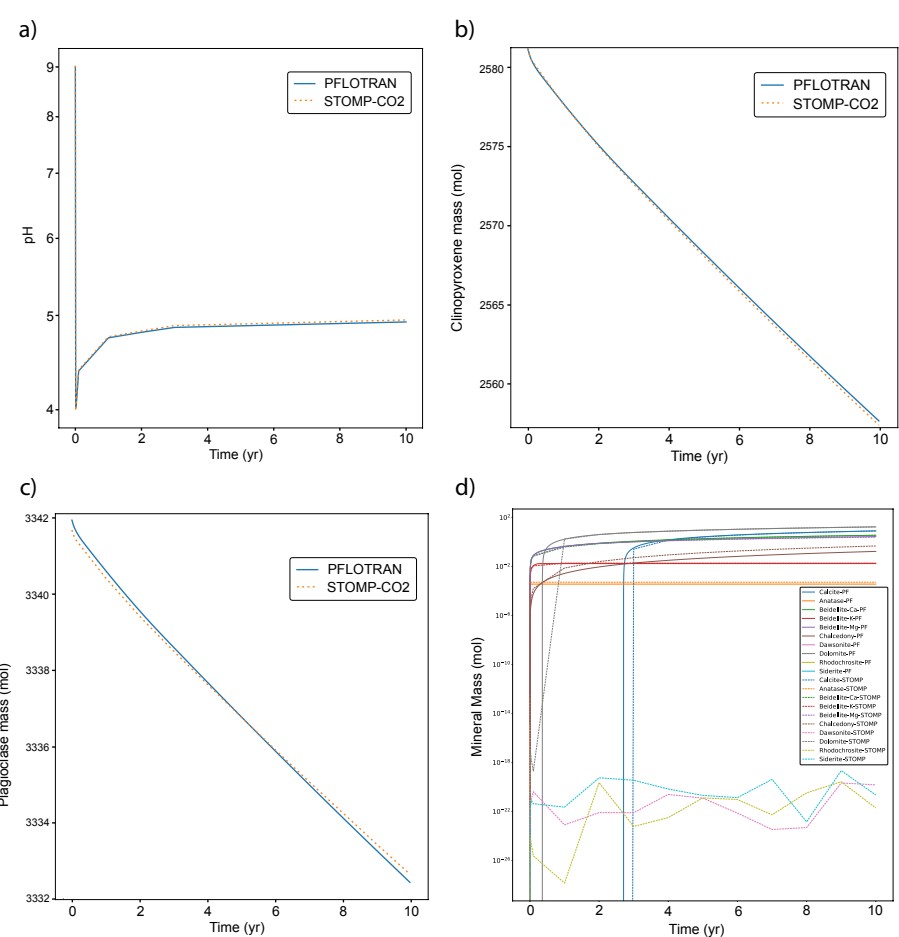

**Figure A3.** Single-cell Reactive $CO_2$ Injection: a) pH, b) Clinopyroxene, c) Plagioclase, and d) Secondary Minerals





## A2    1D Test Cases

Several 1D test cases were built to evaluate SCO2 Mode's representation of boundary conditions, inter-block flux terms, gravity effects, and the fully coupled well model compare to those implemented in STOMP-CO2.

### 955    A2.1    Two-Cell $CO_2$ Injection into a Brine Reservoir

A 2-cell test problem was developed to verify the implementation of interior cell flux terms in SCO2 Mode. In this test, a 1D horizontal Cartesian model with two 15-m long grid cells is initialized to a constant liquid pressure of 12 MPa, zero dissolved $CO_2$ mass fraction, and dissolved salt mass fraction of 0.15 kg/kg. A source term in the left grid cell injects $CO_2$ at 12.5 kg/s, while a boundary condition on the right side of the model holds pressure at the initial pressure. The model is run for $1.0\text{x}10^5$

days. PFLOTRAN and STOMP-CO2 results at several different output times are identical, verifying the implementation of inter-cell multiphase fluxes in SCO2 Mode.



**Figure A4.** Two-Cell $CO_2$ Injection





## A2.2 Two-Cell CO$_2$ Well Injection

In this test, a 2-cell Cartesian model is oriented vertically; each cell has a thickness of 100 m. A fully coupled well model injects CO$_2$ into the domain and is screened through both cells. The well injects CO$_2$ at a total rate of $1.0\text{x}10^6$ kg/year. Since the well is screened through both cells, mass rate of CO$_2$ injected into each cell therefore is distributed to each cell as a function of well pressure, cell pressure, and well index. The model is initialized to a constant liquid pressure of 12 MPa and a salt mass fraction of 0.0475 kg/kg. The top of the model is fixed at the initial liquid pressure and mass fractions; therefore, buoyancy drives upward flow of free-phase CO$_2$ throughout the simulation and the liquid phase sinks to achieve hydrostatic pressure. The model is run to $9\text{x}10^4$ hours. Both the PFLOTRAN and STOMP-CO2 simulations produce identical results at several output times in the simulation, verifying the implementation of the coupled well model in SCO2 Mode.



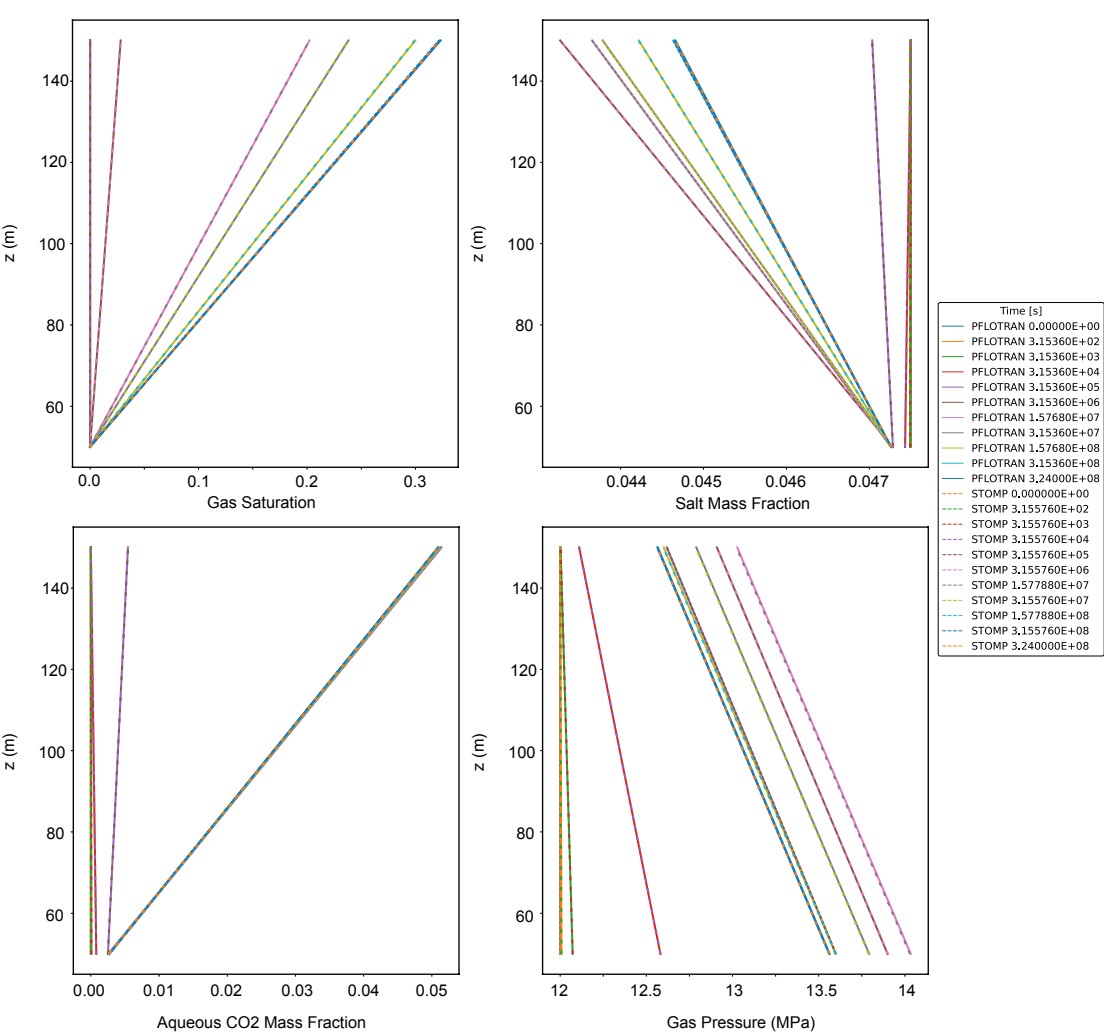

**Figure A5.** Two-Cell $CO_2$ Well Injection





### A2.3   CO$_2$ Injection into a Vertical Brine Reservoir

This test case models a CO$_2$ injection into a 200-m vertical column using a fully coupled well model. Here, each cell is 1 m thick; the vertical well spans the entire domain and is screened throughout. Thus, the well model injects CO$_2$ into many more grid cells than in the previous problem. This problem tests the ability of the well model to inject into hundreds of reservoir grid
cells adaptively depending on bottomhole pressure, reservoir pressure, and well index; additionally it tests the implementation of fluxes across hundreds of reservoir cells. The reservoir is initialized to a constant liquid pressure, zero CO$_2$ mass fraction, and a constant salinity of 0.0475 kg/kg, with a Dirichlet boundary condition at the top set at initial pressure and mass fractions. The model is isothermal at a constant temperature of 20 $^o$C. The model is run for 9x10$^4$ hours. PFLOTRAN and STOMP-CO2 produce nearly identical results at all output times. The one exception is in dissolved CO$_2$ mass fractions at late times in the
deeper part of the model, where both simulators deviate very slightly. These results verify the implementation fully coupled wells in PFLOTRAN where the wells are screened for hundreds of grid cells.





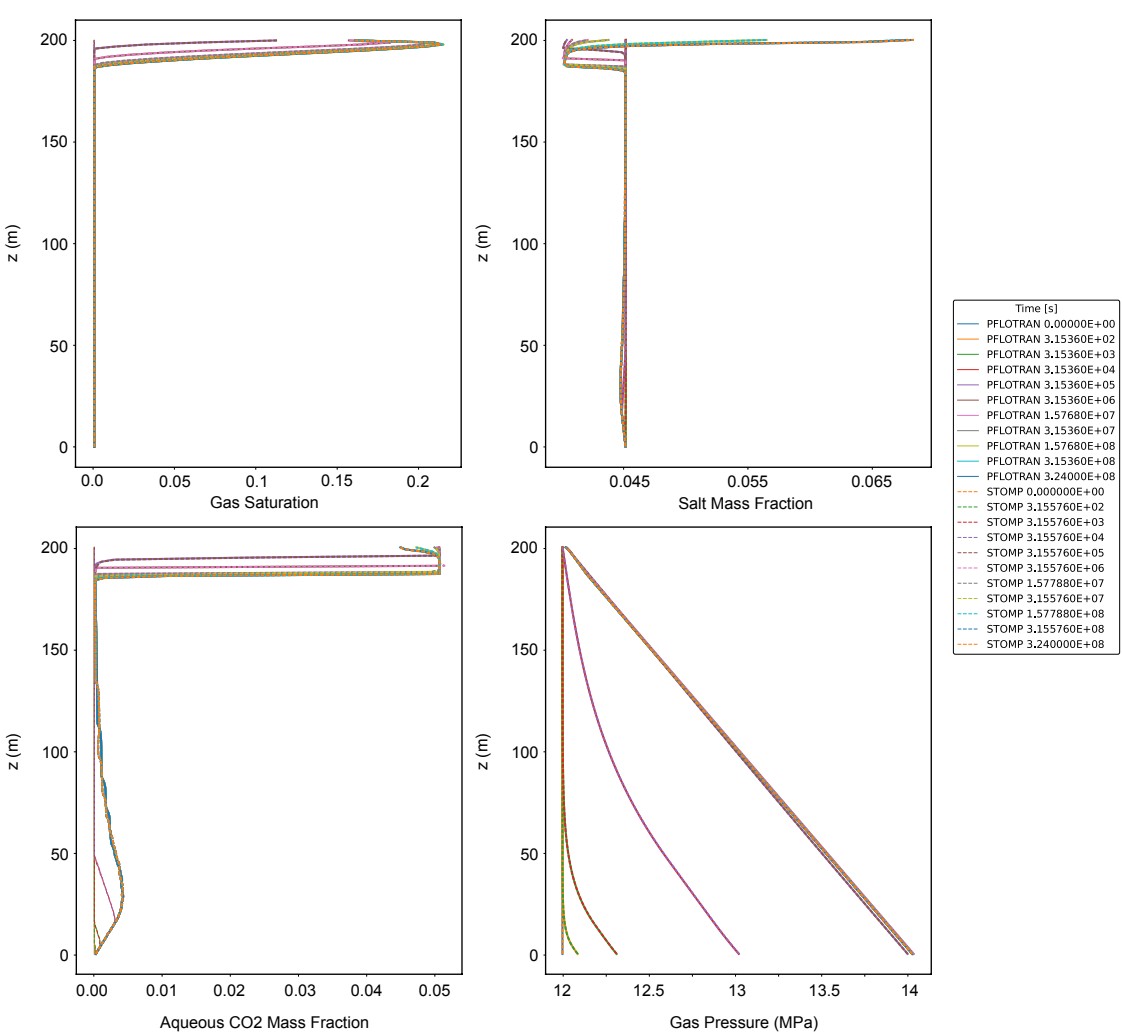

**Figure A6.** Column CO$_2$ Well Injection





*Author contributions.* MN: software development, model conceptualization, formal analysis, methodology, writing; KM: model conceptualization, writing; GH: software development, review, and editing; XH: model conceptualization, writing; PL: model conceptualization, development, and writing

*Competing interests.* The authors declare no competing interests.

*Acknowledgements.* This research was supported by Pacific Northwest National Laboratory's Laboratory-Directed Research and Development (LDRD) program, Award No. 211622. PNNL is operated for the DOE by Battelle Memorial Institute under contract DE-AC05-76RL01830. This paper describes objective technical results and analysis. Any subjective views or opinions that might be expressed in the paper do not necessarily represent the views of the U.S. Department of Energy or the United States Government.





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
