# Peer review of "Modeling Supercritical CO2 Flow and Mineralization in Reactive Host Rocks with PFLOTRAN v7.0"

_EGUsphere, 2025_

## Author Response (AR1)

The authors would like to thank Giacomo Medici and two anonymous reviewers for their constructive feedback. These comments have greatly improved the clarity of the manuscript. We have responded to each reviewer comment here in red.

CC1.

Line 24. I would enlarge the views to the flow and reactive transport of contaminants in geological media in your introduction.

- Thank you. We have added a reference for $CO_2$ as a contaminant, and broadened the description of the PFLOTRAN software.

Line 41. Here, the parallel with flow and transport of contaminants given the fact that you also mention the Darcy and the Fick laws (later in the methodology). Please, insert the parallel with groundwater flow with caprocks vs. aquitard/aquitude, and PFLOTRAN vs. PHREEQC/MT3DMS. See literature below on this parallel where CO2 storage is discussed:

- Medici, G., Munn, J.D., Parker, B.L. 2024. Delineating aquitard characteristics within a Silurian dolostone aquifer using high-density hydraulic head and fracture datasets. Hydrogeology Journal 32, 1663-1691.

- Steel L., Mackay E., Maroto-Valter M.M. 2018. Experimental investigation of CO2-brine-calcite interactions under reservoir conditions. Fuel Processing Technology 169, 122-131.

- Thank you. Darcy flow and Fickian diffusion were tested as part of the benchmarks between PFLOTRAN and STOMP-CO2.

Line 50. Specify the 3 to 4 objectives of your research by using numbers (e.g., i, ii, and iii).

- We have clearly stated the objectives now, thank you.

Line 97. "Advective fluxes", I would also mention laminar flow either here or in the introduction.

- Thank you. This has been added for clarity.

Line 755. Very interesting paragraph. There is a small contrast with part of the title "host rock". Please, provide explanation for this issue.

- This is a 1D vertical simulation meant to test buoyant 1D flow. The "fault" designation is merely to highlight the role of high permeability pathways for potential $CO_2$ leakage.

Figures and tables

Figure 1. Do you need an approximate spatial scale?

- Spatial scales are decided by the modeler. This is a generic schematic of a well model in a discrete system.

Figure 2. Do you need a spatial scale?

- Same as above. Spatial scales are decided by the modeler. This is a generic schematic of a well model in a discrete system.

Figure 7. By contrast, I can see cells with a spatial scale in this figure.

- This is a specific benchmark simulation with defined constraints, so spatial scales are necessary.

Figure 8. Important output. You can move PFLOTRAN and STOMP on the right to gain space, and make the figure larger.

- Thank you. This figure has been updated based off of reviewer responses.

Figure 10. Same here. You can move PFLOTRAN and STOMP on the right to gain space, and make the figure larger.

- Thank you. This figure has been updated based off of reviewer responses.

Figure 11. Do you need a scale bar for your reservoir?

- Thank you for the suggestion. Scale bars have been added.

RC1.

- Review the manuscript for repeated information, especially when describing benchmark problems, equations, or simulation steps. Consolidate where possible. For instance, similar descriptions of PFLOTRAN and STOMP-CO2 verification could be summarized once, with differences highlighted as needed.

  - Thank you. RC2 suggested moving the verification test section into an Appendix. We have moved this section and believe this focuses the bulk of the manuscript with less repetition.

- Preface each benchmark case with a concise statement of its objectives and its relevance to the study.

- o We agree, the motivation got buried in the text and was not completely clear. We have added motivating statements at the beginning of each benchmark description.

- After each benchmark, briefly summarize the findings and their implications before moving on to the next section.

    - o Summarizing conclusions have been added to each benchmark

- The authors should consider using SPE Comparative Study No. 11 to validate STOMP-CO2 outcomes against other group solutions. Although this case excludes geochemical reactions, it would be helpful to test additional capabilities such as mutual solubility, diffusion, and mechanical dispersion.

    - o Thank you. Both PFLOTRAN and STOMP-CO2 were involved in the SPE CSP 11 benchmark.

- In the conclusion, reinforce the main outcomes, their significance, and future directions in a tightly structured paragraph sequence.

    - o Thank you. An additional paragraph has been added that emphasizes the significance of the work and future directions.

Minors:

- Shorten and simplify complex sentences. Several sentences in the manuscript are long and contain multiple clauses, which can make them challenging to follow. Consider breaking these into shorter, more direct sentences.

    - o Thank you. We have cleaned up the wording throughout the manuscript.

- Add transition sentences at the beginning and end of sections to guide readers.

    - o Transition sentences have been added.

RC2.

- Section 2:

- Line 93: how are transitions between gas and liquid CO2-phase properties handed? Particularly in thermal simulations it is possible that CO2 will cross this phase boundary in situ.

    - This is a good point. Within a given grid cell, the $CO_2$-rich phase is treated as a single phase whose properties are defined by the Span-Wagner EOS. So if it is transitioning between liquid and gas, this interface is not explicitly modeled and the $CO_2$-rich phase is modeled with bulk averaged properties. We have updated the language here to be clearer about this limitation.

- Section 2.3.1 Does the formulation for the liquid-phase density account for the increase in density when CO2 is dissolved in brine?

    - Yes.

- Section 2.4.1 line 198 do the capillary pressure extensions depend on salinity to reflect the changes in IFT with salt concentrations? If not, does this create a discrepancy between the capillary pressure extensions and the capillary pressure curves when there are two mobile phases, which do depend on IFT?

    - Yes, $CO_2$-water IFT scaling as a function of salinity is optional using the UPDATE_SURFACE_TENSION keyword. This capability has not been benchmarked and therefore was not included in the test problems in the manuscript. If the user were to invoke IFT scaling, they would need to ensure that capillary pressure measurements were consistent with reference conditions.

- Section 2.5 well model

    - Does the new well model allow for leakage from one completion interval to another in the same well, like T2-well and the PFLOTRAN WIPP-model wells?

        - No, this is a hydrostatic well model and can only simulate either injection or production, not combinations. We have added a clarification to this effect.

    - Line 227 "top hole mass flow rate" this is usually referred to as the surface (mass) flowrate

        - Thank you. We have changed this wording

- - Can the wells be controlled by volume constraint or exclusively using pressure constraints (either surface or downhole?)

    - - Wells are mass rate controlled, with the option to impose minimum or maximum pressures. If wells hit a max/min pressure, they become pressure controlled. There is currently no option to only constrain by surface or downhole pressure, but this could presumably be achieved by setting an unrealistically high rate constraint in addition to a maximum and/or minimum pressure.

  - - Are the pressures enforced at each well-cell in the well trajectory, or is the determination to switch to pressure constraint based on an average pressure?

    - - Fracture pressure and minimum pressure are both imposed on the well bottom hole pressure, the primary variable in the well model. This has been clarified in the manuscript.

- Section 3:

  - I feel like much of the material in Section 3 could be moved to either an appendix or to the online documentation. This is essential information for fellow PFLOTRAN users but adds significantly to the manuscript length. Moving it out of the main manuscript would also make the paper feel less like a user's manual.

    - Thank you. This section has been moved to the Appendix for users to reference.

  - The code snippets would be easier to cross-reference if they were made into figures or tables.

    - Thank you for the suggestion. Input deck snippets have been made into individual tables for easy cross reference

- Section 4

  - Line 765 is the gas-phase boundary condition at the bottom boundary a Dirichlet or Neumann boundary?

    - A Dirichlet BC is imposed. This has been clarified.

  - Table 7: add the depth to the top of the shale layers to the table for this benchmark problem for completeness.

    - Thank you. These have been added to the table.

o Figure 6 and 7 it would be much easier for readers to compare the STOMP/PFLOTRAN simulation results if the same color bar was used for both simulators

- Thank you for the suggestion. The color bars have been updated

o Line 853 how does mineralization impact porosity and/or two-phase flow properties in PFLOTRAN? Is it different than STOMP?

- PFLOTRAN can calculate porosity as a function of mineral volume fractions to account for $CO_2$ mineralization. However, this capability is not employed in this example to be consistent with STOMP's implementation of the benchmarks.